# OBSERVATION-GUIDED DIFFUSION PROBABILISTIC MODELS

## ABSTRACT

We propose a novel diffusion model called observation-guided diffusion probabilistic model (OGDM), which effectively addresses the trade-off between quality control and fast sampling. Our approach reestablishes the training objective by integrating the guidance of the observation process with the Markov chain in a principled way. This is achieved by introducing an additional loss term derived from the observation based on the conditional discriminator on noise level, which employs Bernoulli distribution indicating whether its input lies on the (noisy) real manifold or not. This strategy allows us to optimize the more accurate negative log-likelihood induced in the inference stage especially when the number of function evaluations is limited. The proposed training method is also advantageous even when incorporated only into the fine-tuning process, and it is compatible with various fast inference strategies since our method yields better denoising networks using the exactly same inference procedure without incurring extra computational cost. We demonstrate the effectiveness of the proposed training algorithm using diverse inference methods on strong diffusion model baselines.

## 1 INTRODUCTION

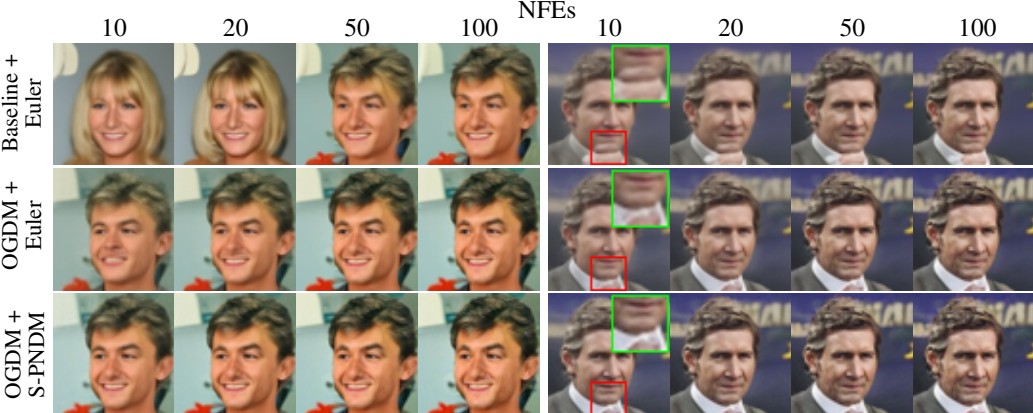

Figure 1: Comparisons of generated images from the same initial noise and deterministic samplers but with different NFEs using the ADM backbone on the CelebA dataset. The entries on the leftmost column of the figure denote the combinations of the training and inference methods. (Left) The baseline model generates samples with inconsistent attributes, *e.g.*, gender, hair, *etc.*, by varying NFEs while our approach preserves such properties. (Right) The samples generated by the baseline method with a small number of NFEs tend to be blurry and unrealistic. Also, they have an unnaturally bright and textureless area around the chin of the person.

Diffusion probabilistic models (Sohl-Dickstein et al., 2015; Ho et al., 2020) have shown impressive generation performance in various domains including image (Rombach et al., 2022; Dhariwal & Nichol, 2021), speech (Kong et al., 2021; Jeong et al., 2021), point cloud (Luo & Hu, 2021), 3D shapes (Zeng et al., 2022), graph (Niu et al., 2020; Hoogeboom et al., 2022), and so on. The key

idea behind these approaches is to formulate data generation as a series of denoising steps of the diffusion process, which sequentially corrupts training data towards a random sample drawn from a prior distribution, *e.g.*, Gaussian distribution.

As diffusion models are trained with an explicit objective, *i.e.*, maximizing log-likelihood, they are advantageous over Generative Adversarial Networks (GANs) (Goodfellow et al., 2014) in terms of learning stability and sample diversity. Moreover, the iterative backward processes and accompanying sampling strategies further improve the quality of samples at the expense of computational efficiency. The tedious inference process involving thousands of network forwarding steps is a critical drawback of diffusion models.

The step size of diffusion models has a significant impact on the expressiveness of the models as the Gaussian assumption imposed on the reverse (denoising) sampling holds only when the step size is sufficiently small (Sohl-Dickstein et al., 2015). On the other hand, the backward distribution deviates from the Gaussian assumption as the step size grows, resulting in an inaccurate modeling. This exacerbates the discrepancy between the training objective and the negative

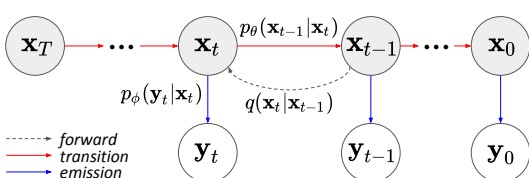

Figure 2: The directed graphical model of the proposed denoising process with observations.

log-likelihood at inference. Consequently, performance degradation is inevitable with coarse time steps. To alleviate the deviation from the true objective caused by large step sizes, our approach incorporates an observation of each state corresponding to perturbed data. To be specific, we consider the data corruption and denoising processes to follow the transition probabilities that respectively align with the forward and backward distributions of Ho et al. (2020) while an observation at each time step following the emission probability aids in achieving a more accurate backward prediction. Figure 2 depicts the directed graphical model of the proposed method.

Our approach offers a significant benefit in the sense that it precisely maximizes the log-likelihood at inference even when employing fast sampling strategies with large step sizes. The observation process plays an important role during training to adjust the denoising steps towards a more accurate data manifold especially when the reverse process deviates from the Gaussian distribution. When it comes to the inference stage, the observation process is no longer an accountable factor and hence incurs no additional computational overhead for sampling.

The main question in this approach is what is observable given a state at each time step. We define an observation following the Bernoulli distribution on the probability of whether noisy data lies on the manifold of real data with the corresponding noise level. From a practical point of view, we implement this observation with a score of a time-dependent discriminator, which takes either true denoised samples or fake ones given by the learned denoising network.

Our main contributions are summarized below:

- We propose an observation-guided diffusion probabilistic model, which accelerates inference speed while maintaining high sample quality. Our approach only alters the training procedures, resulting in no extra computational or memory overhead during inference.

- We derive a principled surrogate loss for the log-likelihood maximization in the observation-guided setting and show its effectiveness in minimizing the KL-divergence between temporally coarse forward and backward processes.

- Our training objective is applicable to various inference methods with proper adjustments, which allows us to utilize various fast sampling strategies that further enhance sample quality.

- The proposed technique can be employed for training from scratch or fine-tuning from pretrained models; compatibility with fine-tuning significantly enhances the practicality of our method.

The rest of this paper is organized as follows. Section 2 reviews related work and Section 3 describes our main algorithm with the justification of the proposed objective. We present experimental results and analyses in Section 4, discuss future work in Section 5, and conclude our paper in Section 6.

## 2 RELATED WORK

There exists a series of studies on diffusion probabilistic models (Sohl-Dickstein et al., 2015; Ho et al., 2020; Song et al., 2021b) that have contributed to accelerated sampling. A simple and intuitive method is to simply skip intermediate time steps and sample a subset of the predefined time steps used for training as suggested by DDIM (Song et al., 2021a). By interpreting the diffusion model as solving a specific SDE (Song et al., 2021b), advanced numerical SDE or ODE solvers (Jolicoeur-Martineau et al., 2021; Dockhorn et al., 2022; Song et al., 2021b; Karras et al., 2022; Liu et al., 2022) are introduced to speed up the backward process. For instance, EDM (Karras et al., 2022) employs a second-order Heun's method (Süli & Mayers, 2003) as its ODE solver, demonstrating that simply adopting existing numerical methods as is can improve performance. On the other hand, some work further refine numerical solvers tailored for diffusion models. For example, PNDM (Liu et al., 2022) provides a pseudo-numerical solver by combining DDIM and high-order classical numerical methods such as Runge-Kutta (Süli & Mayers, 2003) and linear multi-step (Timothy, 2017), and GENIE (Dockhorn et al., 2022) applies a higher-order solver to the DDIM ODE.

On the other hand, Nichol & Dhariwal (2021); Bao et al. (2022a;b); Watson et al. (2022) aim to find the better (optimal) parameters of the reverse process with or without training. For instance, Analytic-DPM (Bao et al., 2022b) presents a training-free inference algorithm by estimating the optimal reverse variances under shortened inference steps and computing the KL-divergence between the corresponding forward and reverse processes in analytic forms. Knowledge distillation (Salimans & Ho, 2022; Luhman & Luhman, 2021; Meng et al., 2023; Song et al., 2023) is another direction for better optimization, where a single time step in a student model learns to simulate the representations from multiple denoising steps in a teacher model. Note that our approach is orthogonal and complementary to the aforementioned studies since our goal is to train better denoising networks robust to inference with large step sizes.

There are a couple of existing methods (Xiao et al., 2022; Wang et al., 2022; Kim et al., 2023) that adopt time-dependent discriminators, yet their motivations and intentions differ significantly from ours. The time-dependent discriminator in DDGAN (Xiao et al., 2022) is designed to guide the generator in approximating non-Gaussian reverse processes while Diffusion-GAN (Wang et al., 2022) employs it to mitigate discriminator overfitting. Kim et al. (2023), on the other hand, utilize the discriminator during inference stages to adjust the score estimation additionally. In contrast, the discriminator in our approach serves as a means to provide observations to diffusion models during the training phase, without involving the inference phase.

## 3 OBSERVATION-GUIDED DIFFUSION PROBABILISTIC MODELS

This section describes the mathematical details of our algorithm and analyzes how to interpret and implement the derived objective function.

### 3.1 PROPERTIES

The proposed observation-guided diffusion probabilistic model, defined by the graphical model in Figure 2, involves two stochastic processes: the state process $\{\mathbf{x}_t\}_{t=0}^T$ and the observation process $\{\mathbf{y}_t\}_{t=0}^T$. The transition and emission probabilities of the forward process, denoted by $q(\mathbf{x}_t|\mathbf{x}_{t-1})$ and $q(\mathbf{y}_t|\mathbf{x}_t)$, respectively, are derived by taking advantage of the following properties given by the graphical model:

$$\mathbf{x}_{t+1}|\mathbf{x}_t \perp\!\!\!\perp \mathbf{x}_{0:t-1}, \mathbf{y}_{0:t} \quad \text{and} \quad \mathbf{y}_t|\mathbf{x}_t \perp\!\!\!\perp \mathbf{x}_{0:t-1}, \mathbf{y}_{0:t-1}, \tag{1}$$

where $\perp\!\!\!\perp$ denotes statistical independence. In the reverse process, we obtain the transition probability, $p(\mathbf{x}_{t-1}|\mathbf{x}_t)$, and the emission probability, $p(\mathbf{y}_t|\mathbf{x}_t)$, using the similar properties as

$$\mathbf{x}_{t-1}|\mathbf{x}_t \perp\!\!\!\perp \mathbf{x}_{T:t+1}, \mathbf{y}_{T:t} \quad \text{and} \quad \mathbf{y}_t|\mathbf{x}_t \perp\!\!\!\perp \mathbf{x}_{T:t+1}, \mathbf{y}_{T:t+1}. \tag{2}$$

### 3.2 NEW SURROGATE OBJECTIVE

Using (1) and Bayes' theorem, we derive the joint probability of the forward process as follows:

$$q(\mathbf{x}_{1:T}, \mathbf{y}_{0:T}|\mathbf{x}_0) = q(\mathbf{x}_T|\mathbf{x}_0) \prod_{t=2}^T q(\mathbf{x}_{t-1}|\mathbf{x}_t, \mathbf{x}_0) \prod_{t=0}^T q(\mathbf{y}_t|\mathbf{x}_t). \tag{3}$$

From (2), the joint probability of the reverse process is given by

$$p(\mathbf{x}_{T:0}, \mathbf{y}_{T:0}) = p(\mathbf{x}_T) \prod_{t=T}^{1} p(\mathbf{x}_{t-1}|\mathbf{x}_t) \prod_{t=T}^{0} p(\mathbf{y}_t|\mathbf{x}_t). \tag{4}$$

Therefore, we derive the upper bound of the expected negative log-likelihood as

$$\mathbb{E}_{\mathbf{x}_0 \sim q}\left[-\log p(\mathbf{x}_0)\right] \tag{5}$$

$$= \mathbb{E}_{\mathbf{x}_0 \sim q}\left[\log \mathbb{E}_{\mathbf{x}_{1:T}, \mathbf{y}_{0:T} \sim q} \frac{q(\mathbf{x}_{1:T}, \mathbf{y}_{0:T}|\mathbf{x}_0)}{p(\mathbf{x}_{0:T}, \mathbf{y}_{0:T})}\right] \tag{6}$$

$$\leq \mathbb{E}_{\mathbf{x}_0 \sim q}\mathbb{E}_{\mathbf{x}_{1:T}, \mathbf{y}_{0:T} \sim q}\left[\log \frac{q(\mathbf{x}_{1:T}, \mathbf{y}_{0:T}|\mathbf{x}_0)}{p(\mathbf{x}_{0:T}, \mathbf{y}_{0:T})}\right] \qquad (\because \text{Jensen's inequality}) \tag{7}$$

$$= \mathbb{E}_{\mathbf{x}_{0:T}, \mathbf{y}_{0:T} \sim q}\left[\log \frac{q(\mathbf{x}_T|\mathbf{x}_0) \prod_{t=2}^{T} q(\mathbf{x}_{t-1}|\mathbf{x}_t, \mathbf{x}_0)}{p(\mathbf{x}_T) \prod_{t=T}^{1} p(\mathbf{x}_{t-1}|\mathbf{x}_t)}\right] + \mathbb{E}_{\mathbf{x}_{0:T}, \mathbf{y}_{0:T} \sim q}\left[\log \frac{\prod_{t=0}^{T} q(\mathbf{y}_t|\mathbf{x}_t)}{\prod_{t=T}^{0} p(\mathbf{y}_t|\mathbf{x}_t)}\right] \tag{8}$$

$$= D_{\text{KL}}(q(\mathbf{x}_T|\mathbf{x}_0)||p(\mathbf{x}_T)) + E_q\left[-\log p(\mathbf{x}_0|\mathbf{x}_1)\right] + \sum_{t=2}^{T} D_{\text{KL}}(q(\mathbf{x}_{t-1}|\mathbf{x}_t, \mathbf{x}_0)||p(\mathbf{x}_{t-1}|\mathbf{x}_t))$$

$$+ \sum_{t=0}^{T} D_{\text{KL}}(q(\mathbf{y}_t|\mathbf{x}_t)||p(\mathbf{y}_t|\mathbf{x}_t)). \tag{9}$$

**Transition probabilities** From Ho et al. (2020), the forward transition probabilities are given by

$$q(\mathbf{x}_0) \coloneqq \text{P}_{\text{data}}(\mathbf{x}_0) \quad \text{and} \quad q(\mathbf{x}_t|\mathbf{x}_{t-1}) \coloneqq \mathcal{N}(\mathbf{x}_t; \sqrt{1 - \beta_t}\mathbf{x}_{t-1}, \beta_t\mathbf{I}), \tag{10}$$

where $\{\beta_t\}_{t=1}^{T}$ are predefined constants. The backward transition probabilities are defined by

$$p_\theta(\mathbf{x}_T) \coloneqq \mathcal{N}(\mathbf{x}_T; \mathbf{0}, \mathbf{I}) \quad \text{and} \quad p_\theta(\mathbf{x}_{t-1}|\mathbf{x}_t) \coloneqq \mathcal{N}\left(\mathbf{x}_{t-1}; \frac{1}{\sqrt{1 - \beta_t}}(\mathbf{x}_t + \beta_t s_\theta(\mathbf{x}_t, t)), \beta_t\mathbf{I}\right), \tag{11}$$

where $s_\theta(\cdot, \cdot)$ denotes a neural network parameterized by $\theta$. Due to the following equation,

$$q(\mathbf{x}_{t-1}|\mathbf{x}_t, \mathbf{x}_0) = \mathcal{N}\left(\mathbf{x}_{t-1}; \frac{1}{\sqrt{1 - \beta_t}}(\mathbf{x}_t + \beta_t \nabla \log q(\mathbf{x}_t|\mathbf{x}_0)), \frac{1 - \bar{\alpha}_{t-1}}{1 - \bar{\alpha}_t}\beta_t\mathbf{I}\right), \tag{12}$$

where $\bar{\alpha}_t = \prod_{s=1}^{t}(1 - \beta_s)$, the first three terms of (9) are optimized by the following loss function:

$$\sum_{t=1}^{T} \lambda_t ||\epsilon - \epsilon_\theta(\sqrt{\bar{\alpha}_t}\mathbf{x}_0 + \sqrt{1 - \bar{\alpha}_t}\epsilon, t)||_2^2 + C, \tag{13}$$

where $\epsilon_\theta(\mathbf{x}_t, t) = \frac{s_\theta(\mathbf{x}_t, t)}{\sqrt{1 - \bar{\alpha}_t}}$ and $C$ is a constant.

**Emission probabilities** We interpret the last term in (9) as an observation about whether the state, $\mathbf{x}_t$, is on the real data manifold or not. Then, the emission probability of the forward and backward processes are defined by Bernoulli distributions as follows:

$$q(\mathbf{y}_t|\mathbf{x}_t) \coloneqq \text{Ber}(1) \quad \text{and} \quad p(\mathbf{y}_t|\mathbf{x}_t) \coloneqq \text{Ber}(D(f(\mathbf{x}_t))), \tag{14}$$

where $f(\cdot)$ is an arbitrary function that projects an input onto a known manifold and $D(\cdot)$ indicates the probability that an input belongs to the manifold of real data. Hence, the KL-divergence of emission, *i.e.*, the last term of (9), is redefined via two different Bernoulli distributions in (14). Eventually, the KL-divergence between two emission distributions is replaced by a log-likelihood of the manifold embedding as follows:

$$\sum_{t=0}^{T} D_{\text{KL}}(q(\mathbf{y}_t|\mathbf{x}_t)||p(\mathbf{y}_t|\mathbf{x}_t)) = \sum_{t=0}^{T} -\log(D(f(\mathbf{x}_t))). \tag{15}$$

### 3.3 MANIFOLD EMBEDDING AND LIKELIHOOD FUNCTION

We now discuss a technically feasible way to implement (15). The only undecided components in (15) are the projection function to a known manifold, $f(\cdot)$, and the likelihood function, $D(\cdot)$.

We define $f(\cdot)$ as a function projecting $\mathbf{x}_t$ onto a manifold of $\mathbf{x}_{t-s} \sim q_{t-s}$ $(t \geq s)$. With the diffusion model, this can be done by running one discretization step of a numerical ODE solver from the noise level of $t$ to $t - s$, denoted by $\Phi(\mathbf{x}_t, t, s; \theta)$. We implement the projection function using the solver as follows:

$$f_\theta(\mathbf{x}_t) := \hat{\mathbf{x}}_{t-s}^\theta = \Phi(\mathbf{x}_t, t, s; \theta). \qquad (16)$$

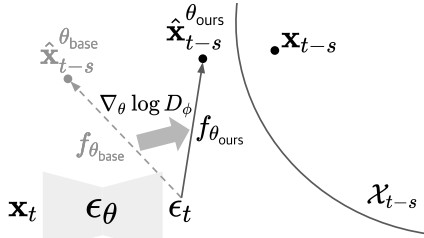

Note that $s$ is a sample drawn from a uniform distribution, $\mathcal{U}(1, \min(t, \lfloor kT \rfloor))$, where $k \in [0, 1]$ is the hyperparameter that determines the lookahead range in the backward direction. We utilize a step of the Euler method [1] or Heun's method [2] to realize the projection function.

On the other hand, $D(\cdot)$ is designed by a discriminator $D_\phi(\cdot)$ to distinguish between projected data from 1) the prediction of the denoising network and 2) real data. Such a design is motivated by the right-hand side of (15), which resembles the objective of the generator in non-saturating GAN (Goodfellow et al., 2014); we employ a time-dependent discriminator taking the projected data, $t$ and $s$, as its inputs.

Figure 3: The role of the discriminator in our objective. $\theta_{\text{ours}}$ and $\theta_{\text{base}}$ denote the denoising parameters learned by the proposed method and the baseline, respectively. The proposed training method nudges the prediction of $\hat{\mathbf{x}}_{t-s}^\theta$ closer to the exact state space than the original.

### 3.4 TRAINING OBJECTIVES

By reformulating the transition and emission probabilities as discussed in Sections 3.2 and 3.3, the first three terms of (9) become (13) while the last term of (9) becomes $-\log(D_\phi(\hat{\mathbf{x}}_{t-s}^\theta, t, s))$. Therefore, the final training objective of the diffusion model with a network design parametrized by $\theta$ is given by

$$\min_\theta \mathbb{E}_{\mathbf{x}_0, \epsilon, t, s} \left[ \underbrace{\lambda_t \|\epsilon - \epsilon_\theta(\sqrt{\bar{\alpha}_t}\mathbf{x}_0 + \sqrt{1 - \bar{\alpha}_t}\epsilon, t)\|_2^2}_{\mathcal{L}_{\text{transition}}} - \gamma \underbrace{\log(D_\phi(\hat{\mathbf{x}}_{t-s}^\theta, t, s))}_{-\mathcal{L}_{\text{emission}}} \right], \qquad (17)$$

where $D_\phi(\cdot)$ denotes a discriminator and $\gamma$ is a hyperparameter.

Besides optimizing $\theta$, we need to train $D_\phi(\cdot)$ to distinguish real data from the prediction of the diffusion model. Following GANs (Goodfellow et al., 2014), the training objective is given by

$$\max_\phi \mathbb{E}_{\mathbf{x}_0, \epsilon, t, s} \left[ \log(D_\phi(\mathbf{x}_{t-s}, t, s)) + \log(1 - D_\phi(\hat{\mathbf{x}}_{t-s}^\theta, t, s)) \right]. \qquad (18)$$

We perform an alternating optimization, where the two objective functions in (17) and (18) take turns for optimization until convergence.

For inference, only diffusion model $\epsilon_\theta$ is taken into account and the discriminator $D_\phi$ is not required. Therefore, no extra computational overhead is imposed when generating samples.

### 3.5 ANALYSIS ON THE OBSERVATION-INDUCED LOSS

We further analyze the surrogate of the negative log-likelihood of a generated sample and explain how the proposed observation-induced loss affects the surrogate.

#### 3.5.1 NEGATIVE LOG-LIKELIHOOD AT INFERENCE

The negative log-likelihood of a generated sample $\mathbf{x}_0 \sim p_\theta$ is given by

$$\mathbb{E}_{\mathbf{x}_0 \sim p_\theta}[-\log q(\mathbf{x}_0)] \leq \sum_{2}^{N} D_{\text{KL}}\left(p_\theta(\mathbf{x}_{\tau_{i-1}}|\mathbf{x}_{\tau_i})||q(\mathbf{x}_{\tau_{i-1}}|\mathbf{x}_{\tau_i})\right) + \mathbb{E}_{p_\theta}[q(\mathbf{x}_{\tau_0}|\mathbf{x}_{\tau_1})], \qquad (19)$$

where $\tau_0 = 0 < \tau_1 < \cdots < \tau_N = T$ is a subsequence of time steps that are selected for fast sampling and $\mathbf{x}_0$ is sampled from $p_\theta$ unlike in (5). Here, $p_\theta(\mathbf{x}_{\tau_{i-1}}|\mathbf{x}_{\tau_i})$ is defined similarly to (11) by replacing $\beta_t$ with $\tilde{\beta}_{\tau_i} = 1 - \frac{\bar{\alpha}_{\tau_i}}{\bar{\alpha}_{\tau_{i-1}}}$.

### 3.5.2 APPROXIMATION ON TRUE REVERSE DISTRIBUTION

While $p_\theta(\mathbf{x}_{\tau_{i-1}}|\mathbf{x}_{\tau_i})$ takes the tractable form of a Gaussian distribution, the true reverse distribution, $q(\mathbf{x}_{\tau_{i-1}}|\mathbf{x}_{\tau_i})$, is still infeasible for estimating the KL-divergence in the right-hand side of (19). To simulate the true reverse density function with $\tilde{\beta}_{\tau_i}$, we use a weighted geometric mean of its asymptotic distributions corresponding to $\tilde{\beta}_{\tau_i} \approx 0$ or 1.

For notational simplicity, let $\mathbf{x}_{\tau_{i-1}} = \mathbf{u}$, $\mathbf{x}_{\tau_t} = \mathbf{v}$, and $\tilde{\beta}_{\tau_i} = \beta$. Then, we denote the true reverse density function as $p_{\mathbf{u}|\mathbf{v}}^{(\beta)}(\mathbf{u}|\mathbf{v})$, for $\mathbf{u} \sim p_\mathbf{u}$ and $\mathbf{v}|\mathbf{u} \sim \mathcal{N}(\sqrt{1-\beta}\mathbf{u}, \beta\mathbf{I})$.

**Lemma 1** *For $\mathbf{u} \sim p_\mathbf{u}$ and $\mathbf{v}|\mathbf{u} \sim \mathcal{N}(\sqrt{1-\beta}\mathbf{u}, \beta\mathbf{I})$, we obtain the following two asymptotic distributions of $p_{\mathbf{u}|\mathbf{v}}^{(\beta)}(\mathbf{u}|\mathbf{v})$:*

$$p_{\mathbf{u}|\mathbf{v}}^{(\beta)}(\mathbf{u}|\mathbf{v}) \approx \mathcal{N}\left(\mathbf{u}; \frac{1}{\sqrt{1-\beta}}(\mathbf{v} + \beta\nabla\log p_\mathbf{u}(\mathbf{v})), \beta\mathbf{I}\right) \text{ for } \beta \ll 1, \tag{20}$$

$$\lim_{\beta\to 1^-} p_{\mathbf{u}|\mathbf{v}}^{(\beta)}(\mathbf{u}|\mathbf{v}) = p_\mathbf{u}(\mathbf{u}). \tag{21}$$

*proof.* Refer to Appendix A.3.

The weighted geometric mean of the two asymptotic distributions in (20) and (21) is given by

$$q_{\mathbf{u}|\mathbf{v}}^{(\xi)}(\mathbf{u}|\mathbf{v}) := C_\xi \, \mathcal{N}\left(\mathbf{u}; \frac{1}{\sqrt{1-\beta}}(\mathbf{v} + \beta\nabla\log p_\mathbf{u}(\mathbf{v})), \beta\mathbf{I}\right)^{1-\xi} p_\mathbf{u}(\mathbf{u})^\xi, \tag{22}$$

where $C_\xi$ is the normalization constant and $\xi$ determines the weight of each component. We further define a mapping function $\xi(\beta)$ that minimizes the difference between $p_{\mathbf{u}|\mathbf{v}}^{(\beta)}(\mathbf{u}|\mathbf{v})$ and $q_{\mathbf{u}|\mathbf{v}}^{(\xi)}(\mathbf{u}|\mathbf{v})$ as

$$\xi(\beta) := \underset{\xi\in[0,1]}{\arg\min} \int_{-\infty}^{\infty} \left(q_{\mathbf{u}|\mathbf{v}}^{(\xi)}(\mathbf{u}|\mathbf{v}) - p_{\mathbf{u}|\mathbf{v}}^{(\beta)}(\mathbf{u}|\mathbf{v})\right)^2 d\mathbf{u}. \tag{23}$$

The exisitence of $\xi(\beta)$ is clear under continuity of $q_{\mathbf{u}|\mathbf{v}}^{(\xi)}(\mathbf{u}|\mathbf{v})$ with respect to $\xi$. For the rest of the analysis, we approximate $p_{\mathbf{u}|\mathbf{v}}^{(\beta)}(\mathbf{u}|\mathbf{v})$ by $q_{\mathbf{u}|\mathbf{v}}^{(\xi(\beta))}(\mathbf{u}|\mathbf{v})$; the validity of the approximation is discussed in Appendix A.4 with more detailed empirical study results. Finally, by substituting the variables back as $\mathbf{u} = \mathbf{x}_{\tau_{i-1}}$, $\mathbf{v} = \mathbf{x}_{\tau_i}$, and $\beta = \tilde{\beta}_{\tau_i}$, the true reverse density function is approximated by

$$q(\mathbf{x}_{\tau_{i-1}}|\mathbf{x}_{\tau_i}) \approx C_{\xi(\tilde{\beta}_{\tau_i})}\mathcal{N}(\mathbf{x}_{\tau_{i-1}}; \boldsymbol{\mu}_{\tau_i}, \tilde{\beta}_{\tau_i}\mathbf{I})^{1-\xi(\tilde{\beta}_{\tau_i})}q(\mathbf{x}_{\tau_{i-1}})^{\xi(\tilde{\beta}_{\tau_i})}, \tag{24}$$

where $\boldsymbol{\mu}_{\tau_i} = \frac{1}{\sqrt{1-\tilde{\beta}_{\tau_i}}}(\mathbf{x}_{\tau_i} + \tilde{\beta}_{\tau_i}\nabla\log q(\mathbf{x}_{\tau_i}))$.

### 3.5.3 INTERPRETATION OF (19)

By using (24), we factorize the KL-divergence term in (19) into a sum of two KL-divergences with a constant as follows:

$$D_{\text{KL}}(p_\theta(\mathbf{x}_{\tau_{i-1}}|\mathbf{x}_{\tau_i})||q(\mathbf{x}_{\tau_{i-1}}|\mathbf{x}_{\tau_i})) + \log C_{\xi(\tilde{\beta}_{\tau_i})}$$

$$\approx (1-\xi(\tilde{\beta}_{\tau_i}))D_{\text{KL}}(p_\theta(\mathbf{x}_{\tau_{i-1}}|\mathbf{x}_{\tau_i})||\mathcal{N}(\mathbf{x}_{\tau_{i-1}}; \boldsymbol{\mu}_{\tau_i}, \tilde{\beta}_{\tau_i}\mathbf{I})) + \xi(\tilde{\beta}_{\tau_i})D_{\text{KL}}(p_\theta(\mathbf{x}_{\tau_{i-1}}|\mathbf{x}_{\tau_i})||q(\mathbf{x}_{\tau_{i-1}}))$$

$$= (1-\xi(\tilde{\beta}_{\tau_i}))||s_\theta(\mathbf{x}_{\tau_i}, \tau_i) - \nabla\log q(\mathbf{x}_{\tau_i})||_2^2 + \xi(\tilde{\beta}_{\tau_i})D_{\text{KL}}(p_\theta(\mathbf{x}_{\tau_{i-1}}|\mathbf{x}_{\tau_i})||q(\mathbf{x}_{\tau_{i-1}})) + C, \tag{25}$$

where $C$ is a constant.

While $\mathcal{L}_{\text{transition}}$ in (17) minimizes the first term of the last equation in (25), $\mathcal{L}_{\text{emission}}$ in (17) minimizes the JS-divergence between two distributions (Goodfellow et al., 2014). Although the JS-divergence has different properties from the KL-divergence, both quantities are minimized when two distributions are equal; the minimization of the JS-divergence effectively reduces the second term of (25) in practice.

Note that the vanilla diffusion models neglect the second term of (25) while DDGAN (Xiao et al., 2022) disregards the first term of (25). On the contrary, the proposed method considers both components, leading to effective optimization. In practice, both $\xi(\tilde{\beta}_{\tau_i})$ and $1 - \xi(\tilde{\beta}_{\tau_i})$ are expected to be non-trivial in fast sampling with relatively large $\tilde{\beta}_{\tau_i}$.

Table 1: FID and recall scores for various NFEs when the projection function $f_\theta(\cdot)$ aligns to the sampler as Euler method [1]. 'OGDM' represents that the models are trained from scratch while 'OGDM (ft)' indicates that the models are fine-tuned from the pretrained baseline models.

| | NFEs | 25 | | 20 | | 15 | | 10 | |
|---|---|---|---|---|---|---|---|---|---|
| Dataset (Backbone) | Method | FID↓ | Rec.↑ | FID↓ | Rec.↑ | FID↓ | Rec.↑ | FID↓ | Rec.↑ |
| CIFAR-10 (ADM) | Baseline | 7.08 | 0.583 | 8.05 | 0.582 | 9.93 | 0.567 | 15.20 | 0.527 |
| | OGDM | **6.26** | **0.587** | **6.81** | **0.587** | **7.96** | **0.578** | 11.63 | 0.546 |
| | OGDM (ft) | 6.69 | 0.582 | 7.26 | 0.581 | 8.15 | 0.571 | **11.18** | **0.549** |
| CIFAR-10 (EDM) | Baseline | 5.32 | 0.572 | 6.82 | 0.558 | 10.02 | 0.524 | 19.32 | 0.452 |
| | OGDM (ft) | **3.21** | **0.603** | **3.53** | **0.600** | **4.64** | **0.587** | **9.28** | **0.546** |
| CelebA (ADM) | Baseline | 7.20 | 0.441 | 7.88 | 0.429 | 9.34 | 0.392 | 11.92 | 0.315 |
| | OGDM | **3.80** | 0.541 | **3.94** | 0.534 | 5.06 | 0.502 | 7.91 | 0.451 |
| | OGDM (ft) | 4.61 | **0.576** | 4.61 | **0.571** | **4.80** | **0.552** | **7.04** | **0.504** |
| LSUN Church (LDM) | Baseline | 7.87 | 0.443 | 8.40 | 0.434 | 8.83 | 0.399 | 15.02 | 0.326 |
| | OGDM (ft) | **7.46** | **0.449** | **7.92** | **0.444** | **8.76** | **0.402** | **14.84** | **0.331** |

Table 2: FID and recall scores for various NFEs when the projection function $f_\theta(\cdot)$ aligns to the sampler as Heun's method [2]. 'OGDM (ft)' indicates that the models are fine-tuned from the pretrained baseline models.

| | NFEs | 35 | | 25 | | 19 | | 15 | | 11 | |
|---|---|---|---|---|---|---|---|---|---|---|---|
| Dataset (Backbone) | Method | FID↓ | Rec.↑ | FID↓ | Rec.↑ | FID↓ | Rec.↑ | FID↓ | Rec.↑ | FID↓ | Rec.↑ |
| CIFAR-10 (EDM) | Baseline | **2.07** | 0.618 | 2.19 | 0.616 | 2.73 | 0.616 | 4.48 | 0.604 | 14.71 | 0.536 |
| | OGDM (ft) | 2.15 | **0.620** | **2.17** | **0.622** | **2.56** | **0.620** | **4.21** | **0.619** | **13.54** | **0.589** |

# 4 EXPERIMENTS

## 4.1 DATASETS, IMPLEMENTATION, AND EVALUATION

**Datasets**  We conduct the unconditional image generations on several standard benchmarks with various resolutions—CIFAR-10 (Krizhevsky et al., 2009) (32×32), CelebA (Liu et al., 2015) (64×64) and LSUN Church (Yu et al., 2015) (256×256).

**Architecture**  We apply our method upon three strong baselines, ADM (Dhariwal & Nichol, 2021) [1] on CIFAR-10 and CelebA, EDM (Karras et al., 2022) [2] on CIFAR-10, and LDM (Rombach et al., 2022) [3] on LSUN Church, using their official source codes. For the implementation of the time-dependent discriminator, we mostly follow the architecture proposed in Diffusion-GAN (Wang et al., 2022), which is based on the implementation of StyleGAN2 (Karras et al., 2020) [4] while time indices are injected into the discriminator as in the conditional GAN. The only modification in our implementation is an additional time index, $s$ in (16), which denotes the number of lookahead time steps from the current time index, $t$. The number of additional parameters required for the implementation of our approach is negligible compared to the baseline methods.

**Training**  We use the default hyperparameters and optimization settings provided by the official codes of baseline algorithms for all experiments except for discriminator training. We consistently obtain favorable results with $k \in [0.1, 0.2]$ and $\gamma \in [0.005, 0.025]$ across all datasets and present our choices of the hyperparameters for reproducibility in Tables 5 and 6 of Appendix C.

**Evaluation protocol**  For quantitative evaluation, we measure FID (Heusel et al., 2017) and recall (Kynkäänniemi et al., 2019) using the implementation provided by ADM[1]. To calculate FID, we employ the full training data as a reference set and 50K generated images as an evaluation set. For the recall metric, we utilize 50K images for both reference and generated sets.

---

[1] https://github.com/openai/guided-diffusion

[2] https://github.com/NVlabs/edm

[3] https://github.com/CompVis/latent-diffusion

[4] https://github.com/NVlabs/stylegan2-ada-pytorch

Table 3: FID and recall scores for various NFEs when the projection function $f_\theta(\cdot)$ is a step of Euler method [1] while two different PNDM (Liu et al., 2022) algorithms are used as samplers. 'OGDM' represents that the models are trained from scratch while 'OGDM (ft)' indicates that the models are fine-tuned from the pretrained baseline models.

| Sampler | Dataset (Backbone) | Method | NFEs 25 | | 20 | | 15 | | 10 | |
| | | | FID↓ | Rec.↑ | FID↓ | Rec.↑ | FID↓ | Rec.↑ | FID↓ | Rec.↑ |
|---|---|---|---|---|---|---|---|---|---|---|
| S-PNDM | CIFAR-10 (ADM) | Baseline | 5.31 | 0.601 | 5.95 | 0.596 | 7.09 | 0.577 | 10.32 | 0.548 |
| | | OGDM | **5.03** | **0.611** | **5.09** | **0.605** | **5.58** | **0.601** | **7.54** | **0.582** |
| | CIFAR-10 (EDM) | Baseline | **2.74** | **0.604** | **3.21** | 0.597 | 4.48 | 0.586 | 9.51 | 0.531 |
| | | OGDM (ft) | 3.75 | **0.604** | 3.62 | **0.605** | **3.60** | **0.600** | **4.97** | **0.586** |
| | CelebA (ADM) | Baseline | 3.67 | 0.553 | 4.15 | 0.539 | 5.22 | 0.511 | 7.33 | 0.445 |
| | | OGDM | **2.62** | **0.607** | **2.70** | **0.604** | **2.96** | **0.585** | **4.35** | **0.545** |
| | LSUN Church (LDM) | Baseline | 8.41 | 0.470 | 8.21 | 0.471 | 8.07 | 0.475 | 9.14 | 0.464 |
| | | OGDM (ft) | **7.69** | **0.480** | **7.48** | **0.489** | **7.48** | **0.481** | **8.68** | **0.478** |
| F-PNDM | CIFAR-10 (ADM) | Baseline | **5.17** | 0.609 | **6.19** | 0.600 | 10.55 | 0.535 | - | - |
| | | OGDM | 5.40 | **0.609** | 6.72 | **0.600** | **8.84** | **0.557** | - | - |
| | CelebA (ADM) | Baseline | 3.39 | 0.562 | 4.25 | 0.539 | 7.08 | 0.488 | - | - |
| | | OGDM | **2.81** | **0.615** | **3.10** | **0.609** | **5.14** | **0.576** | - | - |
| | LSUN Church (LDM) | Baseline | 9.04 | 0.474 | 9.10 | 0.483 | 12.75 | 0.493 | - | - |
| | | OGDM (ft) | **8.24** | **0.481** | **8.39** | **0.495** | **11.78** | **0.505** | - | - |

## 4.2 QUANTITATIVE RESULTS

Tables 1 and 2 demonstrate quantitative comparison results when a projection function $f_\theta(\cdot)$ aligns with a sampler well. They show that a proper combination of a projection function and a sampler substantially improves FID and recall in all cases with NFEs $\leq 25$. This observation implies that the proposed method yields a robust denoising network for large step sizes. Table 1 also compares performance between our models trained from scratch and the ones with fine-tuning on 'CIFAR-10 (ADM)' and 'CelebA (ADM)'. We observe that the fine-tuned models exhibit competitive performance with a small fraction (5–10%) of the training iterations when compared to the models optimized through full training. Detailed analysis and comparisons are provided in Table 6 of Appendix C. These findings highlight the practicality and computational efficiency of the proposed method.

Table 3 presents FID and recall scores in the case that the projection function $f_\theta(\cdot)$ is a step of the Euler method while the samplers are either S-PNDM or F-PNDM. Unless the projection function and the sampler align properly, the benefit from our approach is not guaranteed because the observations may be inaccurately projected onto the manifold from the perspective of the sampler. Despite this reasonable concern, both S-PNDM and F-PNDM still achieve great performance gains especially when NFEs are small. Notably, the combination of the proposed method and S-PNDM shows consistent performance improvements. We infer that the models with the Euler projection and S-PNDM harness synergy because the steps of S-PNDM are similar to the Euler method except for the initial step.

Table 4 presents the FIDs of various algorithms combined with fast inference techniques in a wide range of NFEs. The results show that the proposed approach is advantageous when the number of time steps for inference is small.

## 4.3 QUALITATIVE RESULTS

**Comparsions to baselines** We provide qualitative results on CIFAR-10, CelebA, and LSUN Church obtained by a few sampling steps in comparison with the baseline methods in Appendix E. The baseline models often produce blurry samples when utilizing fast inference methods. In contrast, our models generate crispy and clear images as well as show more diverse colors and tones compared to the corresponding baselines. Since we use deterministic generative process, Moreover, on the left-hand side of Figure 1, the baseline model generates face images with inconsistent genders by varying NFEs. On the other hand, our model maintains the information accurately, which is deirable results under deterministic generative process. This is because the additional loss term of our method enables the model to approximate each backward step more accurately, even with coarse discretization.

Table 4: Comparisons of FIDs on CIFAR-10 (32×32) and CelebA (64×64) to various methods. '†' means the values are copied from the literature and '*' means the values are obtained by applying DDIM (Song et al., 2021a) as a sampler. 'OGDM (euler)' and 'OGDM (heun)' employ a step of Euler and Heun's method as a projection function, respectively.

| Method (+ Sampler) \ NFEs | CIFAR-10 | | | | CelebA | | | |
|---|---|---|---|---|---|---|---|---|
| | 25 | 20 | 15 | 10 | 25 | 20 | 15 | 10 |
| DDIM (Song et al., 2021a) †* | - | 6.84 | - | 13.36 | - | 13.73 | - | 17.33 |
| Analytic-DPM (Bao et al., 2022b) †* | 5.81 | - | - | 14.00 | 9.22 | - | - | 15.62 |
| FastDPM (Kong & Ping, 2021) †* | - | 5.05 | - | 9.90 | - | 10.69 | - | 15.31 |
| GENIE (Dockhorn et al., 2022) † | 3.64 | 3.94 | 4.49 | 5.28 | - | - | - | - |
| Watson et al. (2022) † | 4.25 | 4.72 | 5.90 | 7.86 | - | - | - | - |
| CT (Song et al., 2023) | 6.94 | 6.63 | 6.36 | 6.20 | - | - | - | - |
| **ADM Backbone** | | | | | | | | |
| Baseline + Euler [1] | 7.08 | 8.05 | 9.93 | 15.20 | 7.20 | 7.88 | 9.34 | 11.92 |
| Baseline + S-PNDM | 5.31 | 5.95 | 7.09 | 10.32 | 3.67 | 4.15 | 5.22 | 7.33 |
| Baseline + F-PNDM | 5.17 | 6.19 | 10.55 | - | 3.39 | 4.25 | 7.08 | - |
| OGDM (euler) + Euler [1] | 6.26 | 6.81 | 7.96 | 11.18 | 3.80 | 3.94 | 4.80 | 7.04 |
| OGDM (euler) + S-PNDM | 5.03 | 5.09 | 5.58 | 7.54 | **2.62** | **2.70** | **2.96** | **4.35** |
| **EDM Backbone** | | | | | | | | |
| Baseline + Euler [1] | 5.32 | 6.82 | 10.02 | 19.32 | - | - | - | - |
| Baseline + S-PNDM | 2.74 | **3.21** | 4.48 | 9.51 | - | - | - | - |
| Baseline + Heun's [2] | 2.19 | - | 4.48 | - | - | - | - | - |
| OGDM (euler) + Euler [1] | 3.21 | 3.53 | 4.64 | 9.28 | - | - | - | - |
| OGDM (euler) + S-PNDM | 3.75 | 3.62 | **3.60** | **4.97** | - | - | - | - |
| OGDM (huen) + Heun's [2] | **2.17** | - | 4.21 | - | - | - | - | - |

**Nearest neighborhoods**   Figure 5 of Appendix D illustrates the nearest neighbor examples of our generated examples in the training datasets. According to the results, the samples are sufficiently different from the training examples, confirming that our models do not simply memorize data but increase scores properly by improving diversity in generated samples.

## 5   DISCUSSION

Although not explored in this work, there is more room for amplifying the impact of the proposed approach. For example, integrating the lookahead variable $s$ as an additional input to diffusion models may further improve performance. We can also obtain specialized denoising network for the specific sampling steps by leraning corresponding noise levels of manifolds more frequently during training. This can be realized by selecting $s$ adaptively, rather than uniform sampling. Moreover, considering that $\xi(\beta)$ tends to be positively correlated with $\beta$ (Figure 4(b) of Appendix A.4), it would be reasonable to set a hyperparameter $\gamma$ as an increasing function with respect to $1 - \frac{\bar{\alpha}_t}{\bar{\alpha}_{t-s}}$.

While we examine the observations following the Bernoulli distribution and implement them using the discriminators of GANs, it is important to note that there are multiple possibilities for defining and implementing the observation factors. Although we have presented a single approach in our direction, other generalized formulations of the extended surrogate (9) can be explored; the formulations are potentially specific to target tasks, available resources, and measurement methods. Such flexibility in choosing the best approach for a given situation is one of the advantages of the proposed method.

## 6   CONCLUSION

We presented a diffusion probabilistic model that introduces observations into the plain Markov chain of Ho et al. (2020). As a feasible and effective way forward, we have concretized the surrogate loss for negative log-likelihood using observations following the Bernoulli distribution, and integrated GAN training loss by adding a discriminator network that simulates the observation. Our strategy regulates the denoising network to minimize the accurate negative log-likelihood surrogate at inference, thereby increasing robustness in a few steps sampling. As a result, our method facilitates faster inference by mitigating quality degradation. We demonstrated the effectiveness of the proposed method on well-known baseline models and multiple datasets.

**Reproducibility statement**    We provide implementation and evaluation details in Section 4.1, and hyperparameters in Appendix C to facilitate reproduction of the results presented in Section 4.2. We will release the source code.

**Ethics statement**    Deep generative models possess some potentials to be applied in harmful or abusive contexts. While our work involves unconditional image generations using facial datasets, it primarily presents a generic framework based on diffusion models to alleviate the computational burdensome at inference stage. In fact, enhancing the efficiency of inference is expected to yield societal benefits by making diffusion models more practical for real-world applications and promoting resource conservation. Importantly, our algorithm is not inherently related to specific applications associated with ethical issues.

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

APPENDIX

## A DETAILS OF SECTION 3

### A.1 PROOF OF (3)

$$
\begin{aligned}
q(\mathbf{x}_{1:T}, \mathbf{y}_{0:T} | \mathbf{x}_0) &= q(\mathbf{y}_0 | \mathbf{x}_0) \prod_{t=0}^{T-1} q(\mathbf{x}_{t+1} | \mathbf{x}_{0:t}, \mathbf{y}_{0:t}) q(\mathbf{y}_{t+1} | \mathbf{x}_{0:t+1}, \mathbf{y}_{0:t}) \\
&= q(\mathbf{y}_0 | \mathbf{x}_0) \prod_{t=0}^{T-1} q(\mathbf{x}_{t+1} | \mathbf{x}_t) q(\mathbf{y}_{t+1} | \mathbf{x}_{t+1}) \qquad (\because (1)) \\
&= \prod_{t=0}^{T} q(\mathbf{y}_t | \mathbf{x}_t) \prod_{t=0}^{T-1} q(\mathbf{x}_{t+1} | \mathbf{x}_t) \\
&= \prod_{t=0}^{T} q(\mathbf{y}_t | \mathbf{x}_t) \left[ q(\mathbf{x}_1 | \mathbf{x}_0) \prod_{t=1}^{T-1} q(\mathbf{x}_{t+1} | \mathbf{x}_t, \mathbf{x}_0) \right] \qquad (\because (1)) \\
&= \prod_{t=0}^{T} q(\mathbf{y}_t | \mathbf{x}_t) \left[ q(\mathbf{x}_1 | \mathbf{x}_0) \prod_{t=1}^{T-1} \frac{q(\mathbf{x}_{t+1}, \mathbf{x}_t | \mathbf{x}_0)}{q(\mathbf{x}_t | \mathbf{x}_0)} \right] \\
&= \prod_{t=0}^{T} q(\mathbf{y}_t | \mathbf{x}_t) \left[ q(\mathbf{x}_1 | \mathbf{x}_0) \prod_{t=1}^{T-1} \frac{q(\mathbf{x}_{t+1} | \mathbf{x}_0) q(\mathbf{x}_t | \mathbf{x}_{t+1}, \mathbf{x}_0)}{q(\mathbf{x}_t | \mathbf{x}_0)} \right] \\
&= q(\mathbf{x}_T | \mathbf{x}_0) \prod_{t=2}^{T} q(\mathbf{x}_{t-1} | \mathbf{x}_t, \mathbf{x}_0) \prod_{t=0}^{T} q(\mathbf{y}_t | \mathbf{x}_t).
\end{aligned}
$$

### A.2 PROOF OF (4)

$$
\begin{aligned}
p(\mathbf{x}_{T:0}, \mathbf{y}_{T:0}) &= p(\mathbf{x}_T) p(\mathbf{y}_T | \mathbf{x}_T) \prod_{t=T}^{1} p(\mathbf{x}_{t-1} | \mathbf{x}_{T:t}, \mathbf{y}_{T:t}) p(\mathbf{y}_{t-1} | \mathbf{x}_{T:t-1}, \mathbf{y}_{T:t}) \\
&= p(\mathbf{x}_T) p(\mathbf{y}_T | \mathbf{x}_T) \prod_{t=T}^{1} p(\mathbf{x}_{t-1} | \mathbf{x}_t) p(\mathbf{y}_{t-1} | \mathbf{x}_{t-1}) (\because (1)) \\
&= p(\mathbf{x}_T) \prod_{t=T}^{1} p(\mathbf{x}_{t-1} | \mathbf{x}_t) \prod_{t=T}^{0} p(\mathbf{y}_t | \mathbf{x}_t).
\end{aligned}
$$

### A.3 PROOF OF LEMMA 1

**Lemma 1** *For* $\mathbf{u} \sim p_{\mathbf{u}}$ *and* $\mathbf{v} | \mathbf{u} \sim \mathcal{N}(\sqrt{1 - \beta}\mathbf{u}, \beta\mathbf{I})$, *we obtain the following two asymptotic distributions of* $p_{\mathbf{u} | \mathbf{v}}^{(\beta)}(\mathbf{u} | \mathbf{v})$:

$$
p_{\mathbf{u} | \mathbf{v}}^{(\beta)}(\mathbf{u} | \mathbf{v}) \approx \mathcal{N}\left( \mathbf{u}; \frac{1}{\sqrt{1 - \beta}}(\mathbf{v} + \beta \nabla \log p_{\mathbf{u}}(\mathbf{v})), \beta\mathbf{I} \right) \text{ for } \beta \ll 1, \tag{20}
$$

$$
\lim_{\beta \to 1^-} p_{\mathbf{u} | \mathbf{v}}^{(\beta)}(\mathbf{u} | \mathbf{v}) = p_{\mathbf{u}}(\mathbf{u}). \tag{21}
$$

***proof of (20).*** Let $\mathbf{u}' = \sqrt{1 - \beta}\mathbf{u}$. Then,

$$
p_{\mathbf{v} | \mathbf{u}'}(\mathbf{v} | \mathbf{u}') = \mathcal{N}(\mathbf{v}; \mathbf{u}', \beta\mathbf{I}) = (2\pi\beta)^{-d/2} \exp(-\frac{1}{2\beta} ||\mathbf{v} - \mathbf{u}'||^2) = \mathcal{N}(\mathbf{u}'; \mathbf{v}, \beta\mathbf{I}) = q_{\mathbf{u}' | \mathbf{v}}(\mathbf{u}' | \mathbf{v})
$$

By talyor expansion of $p_{\mathbf{u}'}(\mathbf{u}')$ at $\mathbf{v}$,

$$
p_{\mathbf{u}'}(\mathbf{u}') = p_{\mathbf{u}'}(\mathbf{v}) + < \nabla p_{\mathbf{u}'}(\mathbf{v}), \mathbf{u}' - \mathbf{v} > + O(||\mathbf{u}' - \mathbf{v}||^2).
$$

Then,

$$\int q_{\mathbf{u}'|\mathbf{v}}(\mathbf{u}'|\mathbf{v})p_{\mathbf{u}'}(\mathbf{u}')d\mathbf{u}' = \mathbb{E}_{\mathbf{u}'\sim\mathcal{N}(\mathbf{v},\beta\mathbf{I})}[p_{\mathbf{u}'}(\mathbf{v})+<\nabla p_{\mathbf{u}'}(\mathbf{v}),\mathbf{u}'-\mathbf{v}>+O(||\mathbf{u}'-\mathbf{v}||^2)]$$

$$= p_{\mathbf{u}'}(\mathbf{v}) + O(\beta).$$

By Bayes' rule and above result,

$$p_{\mathbf{u}'|\mathbf{v}}(\mathbf{u}'|\mathbf{v}) = \frac{p_{\mathbf{v}|\mathbf{u}'}(\mathbf{v}|\mathbf{u}')p_{\mathbf{u}'}(\mathbf{u}')}{\int p_{\mathbf{v}|\mathbf{u}'}(\mathbf{v}|\mathbf{u}')p_{\mathbf{u}'}(\mathbf{u}')d\mathbf{u}'}$$

$$= \frac{q_{\mathbf{u}'|\mathbf{v}}(\mathbf{u}'|\mathbf{v})p_{\mathbf{u}'}(\mathbf{u}')}{\int q_{\mathbf{u}'|\mathbf{v}}(\mathbf{u}'|\mathbf{v})p_{\mathbf{u}'}(\mathbf{u}')d\mathbf{u}'}$$

$$= q_{\mathbf{u}'|\mathbf{v}}(\mathbf{u}'|\mathbf{v})\frac{p_{\mathbf{u}'}(\mathbf{v})+<\nabla p_{\mathbf{u}'}(\mathbf{v}),\mathbf{u}'-\mathbf{v}>+O(||\mathbf{u}'-\mathbf{v}||^2)}{p_{\mathbf{u}'}(\mathbf{v})+O(\beta)}$$

$$= q_{\mathbf{u}'|\mathbf{v}}(\mathbf{u}'|\mathbf{v})(1+<\frac{\nabla p_{\mathbf{u}'}(\mathbf{v})}{p_{\mathbf{u}'}(\mathbf{v})},\mathbf{u}'-\mathbf{v}>+O(||\mathbf{u}'-\mathbf{v}||^2))(1+O(\beta))$$

$$= q_{\mathbf{u}'|\mathbf{v}}(\mathbf{u}'|\mathbf{v})\exp(<\nabla\log p_{\mathbf{u}'}(\mathbf{v}),\mathbf{u}'-\mathbf{v}>)+O(\beta)$$

$$= (2\pi\beta)^{-d/2}\exp(-\frac{1}{2\beta}||\mathbf{v}-\mathbf{u}'||^2)\exp(<\nabla\log p_{\mathbf{u}'}(\mathbf{v}),\mathbf{u}'-\mathbf{v}>)+O(\beta)$$

$$= (2\pi\beta)^{-d/2}\exp(-\frac{1}{2\beta}(||\mathbf{v}-\mathbf{u}'||^2-2\beta<\nabla\log p_{\mathbf{u}'}(\mathbf{v}),\mathbf{u}'-\mathbf{v}>))+O(\beta)$$

$$= (2\pi\beta)^{-d/2}\exp(-\frac{1}{2\beta}||\mathbf{u}'-\mathbf{v}-\beta\nabla\log p_{\mathbf{u}'}(\mathbf{v})||^2+O(\beta))+O(\beta)$$

$$\approx \mathcal{N}(\mathbf{u}';\mathbf{v}+\beta\nabla\log p_{\mathbf{u}'}(\mathbf{v}),\beta\mathbf{I}) \text{ for } \beta \ll 1.$$

From $\mathbf{u}' = \sqrt{1-\beta}\mathbf{u}$,

$$p_{\mathbf{u}|\mathbf{v}}(\mathbf{u}|\mathbf{v}) = \mathcal{N}(\mathbf{u};\frac{1}{\sqrt{1-\beta}}(\mathbf{v}+\beta\nabla\log p_{\mathbf{u}}(\mathbf{v})),\frac{\beta}{1-\beta}\mathbf{I})$$

$$\approx \mathcal{N}(\mathbf{u};\frac{1}{\sqrt{1-\beta}}(\mathbf{v}+\beta\nabla\log p_{\mathbf{u}}(\mathbf{v})),\beta\mathbf{I}) \text{ for } \beta \ll 1\blacksquare$$

***proof of (21).***

$$\lim_{\beta\to1^-}p_{\mathbf{v}|\mathbf{u}}(\mathbf{v}|\mathbf{u}) = \lim_{\beta\to1^-}(2\pi\beta)^{-d/2}\exp(-\frac{1}{2\beta}||\mathbf{v}-\sqrt{1-\beta}\mathbf{u}||^2)$$

$$= (2\pi)^{-d/2}\exp(-\frac{1}{2}||\mathbf{v}||^2) := f(\mathbf{v}).$$

From Bayes' rule and above results,

$$\therefore \lim_{\beta\to1^-}p_{\mathbf{u}|\mathbf{v}}(\mathbf{u}|\mathbf{v}) = \lim_{\beta\to1^-}\frac{p_{\mathbf{v}|\mathbf{u}}(\mathbf{v}|\mathbf{u})p_{\mathbf{u}}(\mathbf{u})}{\int p_{\mathbf{v}|\mathbf{u}}(\mathbf{v}|\mathbf{u})p_{\mathbf{u}}(\mathbf{u})d\mathbf{u}}$$

$$= \frac{\lim_{\beta\to1^-}p_{\mathbf{v}|\mathbf{u}}(\mathbf{v}|\mathbf{u})p_{\mathbf{u}}(\mathbf{u})}{\int\lim_{\beta\to1^-}p_{\mathbf{v}|\mathbf{u}}(\mathbf{v}|\mathbf{u})p_{\mathbf{u}}(\mathbf{u})d\mathbf{u}}$$

$$= \frac{f(\mathbf{v})p_{\mathbf{u}}(\mathbf{u})}{\int f(\mathbf{v})p_{\mathbf{u}}(\mathbf{u})d\mathbf{u}}$$

$$= \frac{p_{\mathbf{u}}(\mathbf{u})}{\int p_{\mathbf{u}}(\mathbf{u})d\mathbf{u}} = p_{\mathbf{u}}(\mathbf{u})\blacksquare$$

## A.4   BEHAVIORS OF $p_{\mathbf{u}|\mathbf{v}}^{(\beta)}(\mathbf{u}|\mathbf{v})$, $q_{\mathbf{u}|\mathbf{v}}^{(\xi)}(\mathbf{u}|\mathbf{v})$, AND $\xi(\beta)$.

In this section, we justify the approximation on the density function of reverse distribution by showing behaviors of $p_{\mathbf{u}|\mathbf{v}}^{(\beta)}(\mathbf{u}|\mathbf{v})$, $q_{\mathbf{u}|\mathbf{v}}^{(\xi)}(\mathbf{u}|\mathbf{v})$, and $\xi(\beta)$ on toy examples.

Followings are the definitions in Sections 3.5. For $\mathbf{u} \sim p_{\mathbf{u}}$ and $\mathbf{v}|\mathbf{u} \sim \mathcal{N}(\sqrt{1-\beta}\mathbf{u}, \beta\mathbf{I})$, $p_{\mathbf{u}|\mathbf{v}}^{(\beta)}(\mathbf{u}|\mathbf{v})$ is a real backward density function,

$$q_{\mathbf{u}|\mathbf{v}}^{(\xi)}(\mathbf{u}|\mathbf{v}) = C(\mathcal{N}(\mathbf{u}; \frac{1}{\sqrt{1-\beta}}(\mathbf{v} + \beta\nabla \log p_{\mathbf{u}}(\mathbf{v})), \beta\mathbf{I}))^{1-\xi} p(\mathbf{u})^{\xi},$$

and

$$\xi(\beta) \in \underset{\xi \in [0,1]}{\arg\min} \int_{-\infty}^{\infty} (q_{\mathbf{u}|\mathbf{v}}^{(\xi)}(\mathbf{u}|\mathbf{v}) - p_{\mathbf{u}|\mathbf{v}}^{(\beta)}(\mathbf{u}|\mathbf{v}))^2 d\mathbf{u}.$$

For arbitrary $\mu > 0$, let

$$p_{\mathbf{u}}(\mathbf{u}) = \frac{1}{2}(\mathcal{N}(\mathbf{u}; \mu, 1) + \mathcal{N}(\mathbf{u}; -\mu, 1))$$

$$= \frac{1}{2}(2\pi)^{-1/2}(\exp(-(\mathbf{u}-\mu)^2/2) + \exp(-(\mathbf{u}+\mu)^2/2))$$

Then, $\mathcal{N}(\mathbf{u}; \frac{1}{\sqrt{1-\beta}}(\mathbf{v} + \beta\nabla \log p_{\mathbf{u}}(\mathbf{v})), \beta)$, and $p_{\mathbf{u}|\mathbf{v}}^{(\beta)}(\mathbf{u}|\mathbf{v}) = \dfrac{p_{\mathbf{v}|\mathbf{u}}(\mathbf{v}|\mathbf{u})p_{\mathbf{u}}(\mathbf{u})}{\int p_{\mathbf{v}|\mathbf{u}}(\mathbf{v}|\mathbf{u})p_{\mathbf{u}}(\mathbf{u})d\mathbf{u}}$ can be explicitly expressed. Moreover, $q_{\mathbf{u}|\mathbf{v}}^{(\xi)}(\mathbf{u}|\mathbf{v})$ can be calculated numerically. Therefore, we can numerically calculate $\ell^2$-norm between $p_{\mathbf{u}|\mathbf{v}}^{(\beta)}(\mathbf{u}|\mathbf{v})$ and $q_{\mathbf{u}|\mathbf{v}}^{(\xi)}(\mathbf{u}|\mathbf{v})$. Finally, we can obtain $\xi(\beta)$ for each $\mathbf{v}$ and $\beta$.

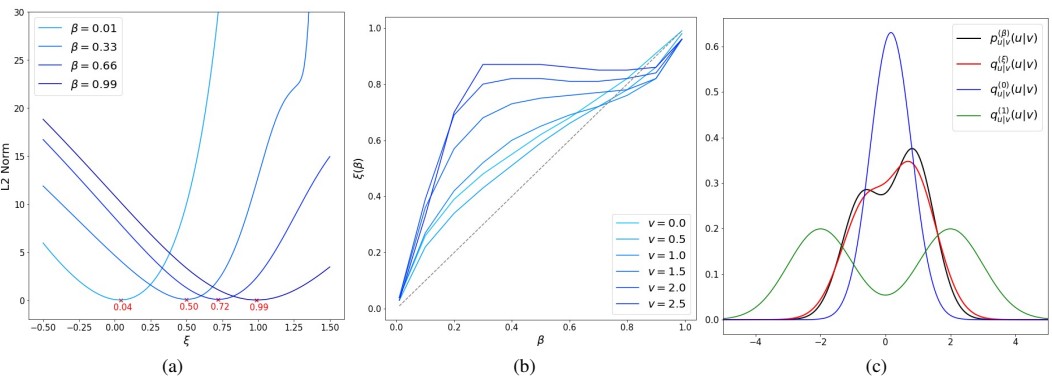

(a)  (b)  (c)

Figure 4: Simulations when $\mu = 2$. (a) $\ell^2$-norm between $p_{\mathbf{u}|\mathbf{v}}^{(\beta)}(\mathbf{u}|\mathbf{v})$, and $q_{\mathbf{u}|\mathbf{v}}^{(\xi)}(\mathbf{u}|\mathbf{v})$ with respect to $\xi$ when $\mathbf{v} = 0.1$. (b) The graph of $\xi(\beta)$ with respect to $\beta$ for various $\mathbf{v}$. (c) Pdfs of four distributions when $\mathbf{v} = 0.1$, $\beta = 0.4$, and $\xi = \xi(\beta) = 0.55$.

Figure 4(a) shows the that $q_{\mathbf{u}|\mathbf{v}}^{(\xi(\beta))}(\mathbf{u}|\mathbf{v})$ better approximates $p_{\mathbf{u}|\mathbf{v}}^{(\beta)}(\mathbf{u}|\mathbf{v})$ than $q_{\mathbf{u}|\mathbf{v}}^{(1)}(\mathbf{u}|\mathbf{v}) = p_{\mathbf{u}}(\mathbf{u})$ and $q_{\mathbf{u}|\mathbf{v}}^{(0)}(\mathbf{u}|\mathbf{v}) = \mathcal{N}(\mathbf{u}; \frac{1}{\sqrt{1-\beta}}(\mathbf{v} + \beta\nabla \log p_{\mathbf{u}}(\mathbf{v})), \beta)$. We can observe that $0 < \xi(\beta) < 1$ for $\forall \beta \in (0,1)$ from Figure 4(b). In Figure 4(c), the black line is the precise reverse distribution while the blue line is asymptotic function when $\beta \to 0^+$ and the green line is the limit when $\beta \to 1^-$. Note that the blue line is the approximation used in vanilla diffusion models. The red line which is a normalized weighted geometric mean of blue and green lines better approximates the real distribution.

## B  NUMERICAL ODE SOLVERS

Suppose we are calculating numerical solution for ODE at $t = 0$ with initial condition $\mathbf{x}_T$ for

$$d\mathbf{x}_t = f(\mathbf{x}_t, t)dt. \tag{26}$$

---

**Algorithm 1:** Euler Method

---

$T = t_N > t_{k-1} > \cdots > t_0 = 0$
**for** $i = N, \cdots, 1$ **do**
  $\quad \mathbf{x}_{t_{i-1}} = \mathbf{x}_{t_i} + (t_{i-1} - t_i)f(\mathbf{x}_{t_i}, t_i)$
**end**
**return** $\mathbf{x}_0$

---

**Algorithm 2:** Heun's Method

---

$T = t_N > t_{k-1} > \cdots > t_0 = 0$
**for** $i = N, \cdots, 1$ **do**
  $\quad \mathbf{x}_{t_{i-1}} = \mathbf{x}_{t_i} + (t_{i-1} - t_i)f(\mathbf{x}_{t_i}, t_i)$
  $\quad$**if** $i > 1$ **then**
    $\quad\quad \mathbf{x}_{t_{i-1}} = \mathbf{x}_{t_i} + (t_{i-1} - t_i)(\frac{1}{2}f(\mathbf{x}_{t_i}, t_i) + \frac{1}{2}f(\mathbf{x}_{t_{i-1}}, t_{i-1}))$
  $\quad$**end**
**end**
**return** $\mathbf{x}_0$

---

## C  HYPERPARAMETERS FOR EXPERIMENTS

Table 5: Hyperparameters for Training

| Dataset | Backbone | Training | Projection | $k$ | $\gamma$ | Batch size | Seen Images |
|---|---|---|---|---|---|---|---|
| CIFAR-10 | ADM | Baseline | - | - | - | 128 | 38M |
| | | OGDM | Euler | 0.1 | 0.01 | 128 | 38M |
| | | OGDM (ft) | Euler | 0.2 | 0.025 | 128 | 2M |
| | EDM | Baseline | - | - | - | 512 | 200M |
| | | OGDM (ft) | Euler | 0.2 | 0.025 | 512 | 20M |
| | | OGDM (ft) | Heun's | 0.2 | 0.005 | 512 | 20M |
| CelebA | ADM | Baseline | - | - | - | 128 | 38M |
| | | OGDM | Euler | 0.1 | 0.01 | 128 | 38M |
| | | OGDM (ft) | Euler | 0.2 | 0.025 | 128 | 2M |
| LSUN Church | LDM | Baseline | - | - | - | 96 | 48M |
| | | OGDM (ft) | Euler | 0.1 | 0.01 | 96 | 1.5M |

Table 6: Hyperparameters for Sampling

| Dataset | Backbone | Sampler | Discretization |
|---|---|---|---|
| CIFAR-10 | ADM | Euler | quadratic |
| | | S-PNDM | linear |
| | | F-PNDM | linear |
| | EDM | Euler | default of Karras et al. (2022) |
| | | Heun's | default of Karras et al. (2022) |
| | | S-PNDM | default of Karras et al. (2022) |
| CelebA | ADM | Euler | linear |
| | | S-PNDM | linear |
| | | F-PNDM | linear |
| LSUN Church | LDM | Euler | linear |
| | | S-PNDM | quadratic |
| | | F-PNDM | quadratic |

## D    NEAREST NEIGHBORHOODS

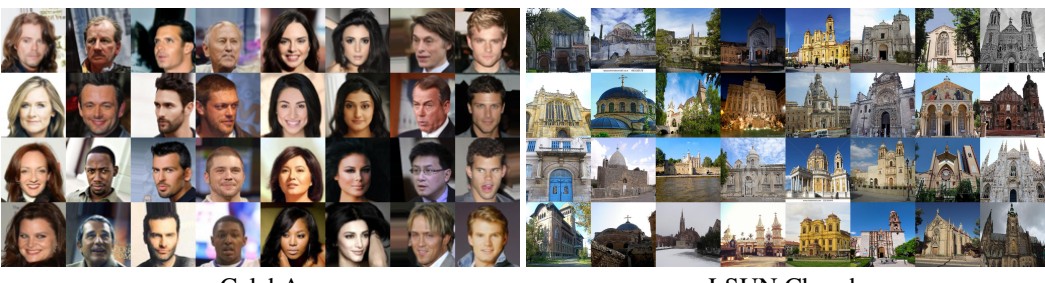

CelebA                                    LSUN Church

Figure 5: Nearest neighborhoods of generated samples from CelebA and LSUN Church datasets. The top row showcases our generated samples using DDIM-50 sampler, while the remaining three rows display the nearest neighborhoods from each training dataset. The distances are measured in the Inception-v3 (Szegedy et al., 2016) feature space.

## E    QUALITATIVE COMPARISONS

We compare the generated images between the baseline and our method using few number of NFEs in Figures 6–21. While we use Euler method and PNDM for sampling in common, for CIFAR-10, we further compare the results on EDM backbone sampled by Heun's method. The images generated by our method have more vivid color and clearer and less prone to produce unrealistic samples compared to the baselines. Also, our method complements advanced samplers, other than Euler method, effectively.

## E.1 CIFAR-10 SAMPLES WITH ADM BASELINE

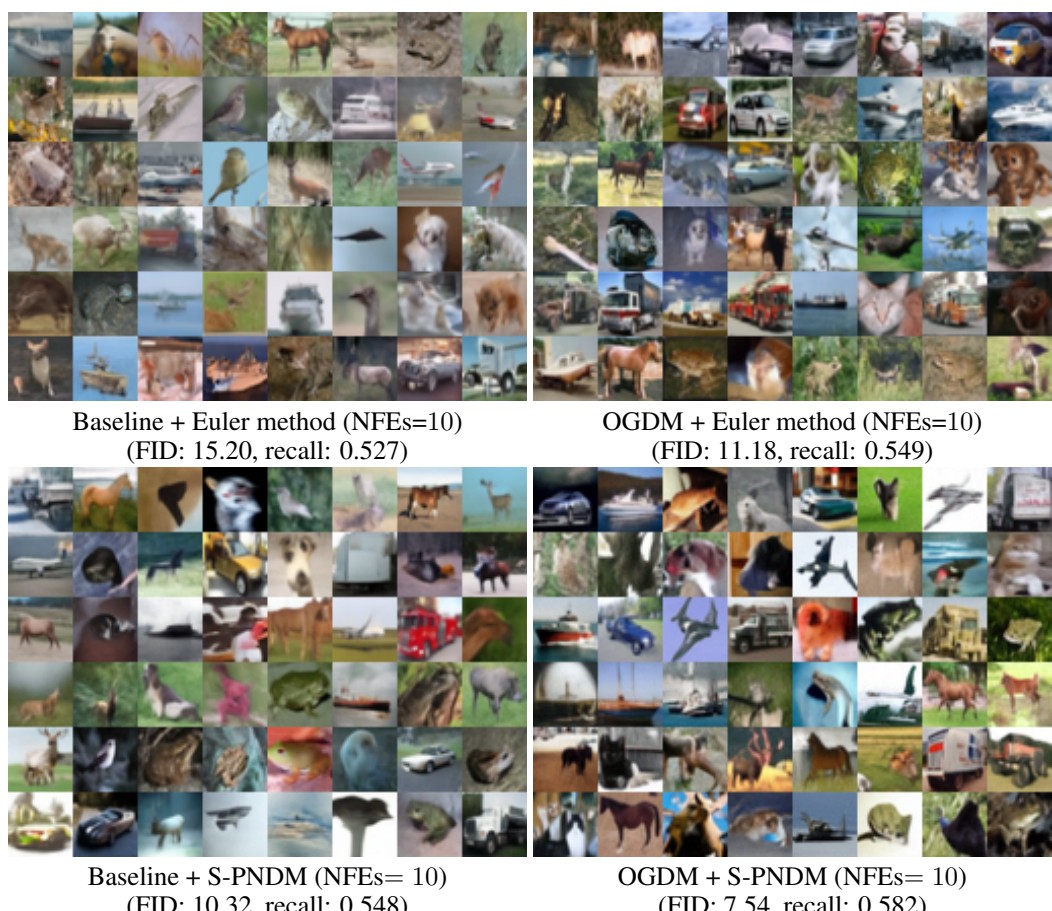

| Baseline + Euler method (NFEs=10) | OGDM + Euler method (NFEs=10) |
|---|---|
| (FID: 15.20, recall: 0.527) | (FID: 11.18, recall: 0.549) |

| Baseline + S-PNDM (NFEs= 10) | OGDM + S-PNDM (NFEs= 10) |
|---|---|
| (FID: 10.32, recall: 0.548) | (FID: 7.54, recall: 0.582) |

Figure 6: Qualitative results on CIFAR-10 dataset with the ADM backbone using Euler method (top) and S-PNDM (bottom) with NFEs=10.

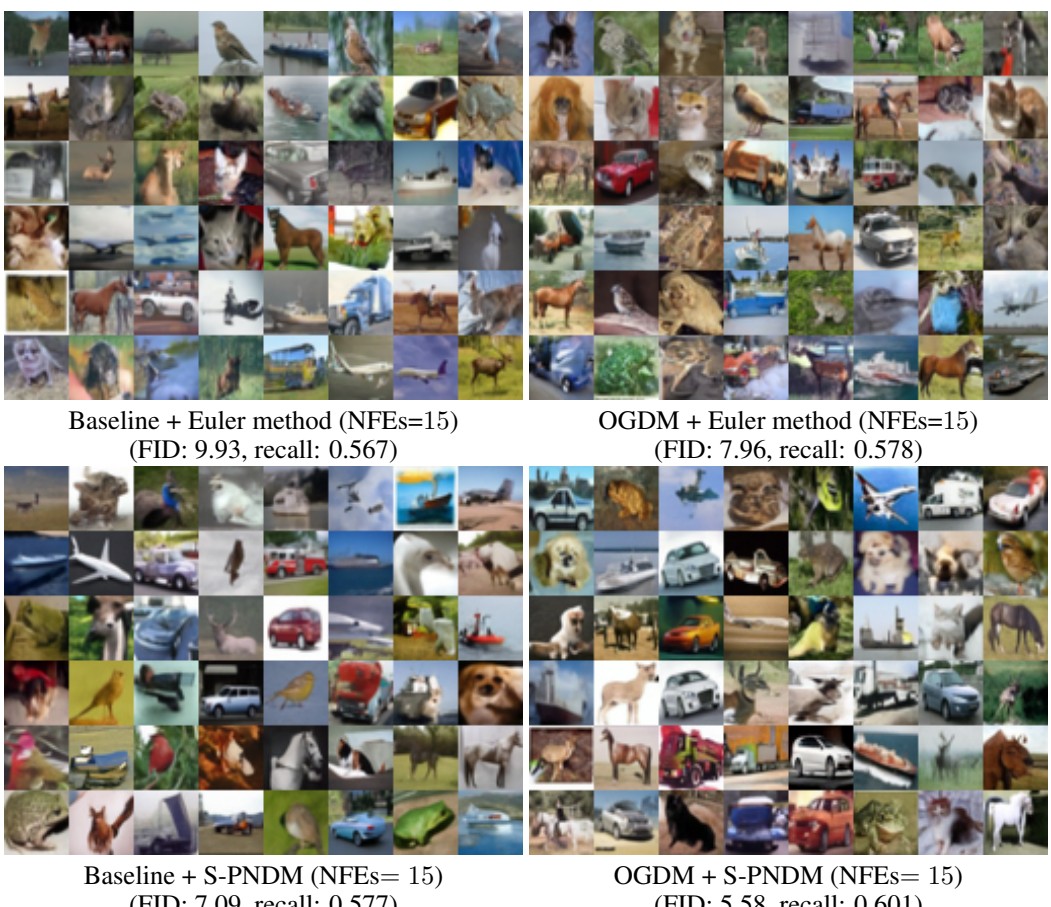

Baseline + Euler method (NFEs=15)
(FID: 9.93, recall: 0.567)

OGDM + Euler method (NFEs=15)
(FID: 7.96, recall: 0.578)

Baseline + S-PNDM (NFEs= 15)
(FID: 7.09, recall: 0.577)

OGDM + S-PNDM (NFEs= 15)
(FID: 5.58, recall: 0.601)

Figure 7: Qualitative results on CIFAR-10 dataset with the ADM backbone using Euler method (top) and S-PNDM (bottom) with NFEs=15.

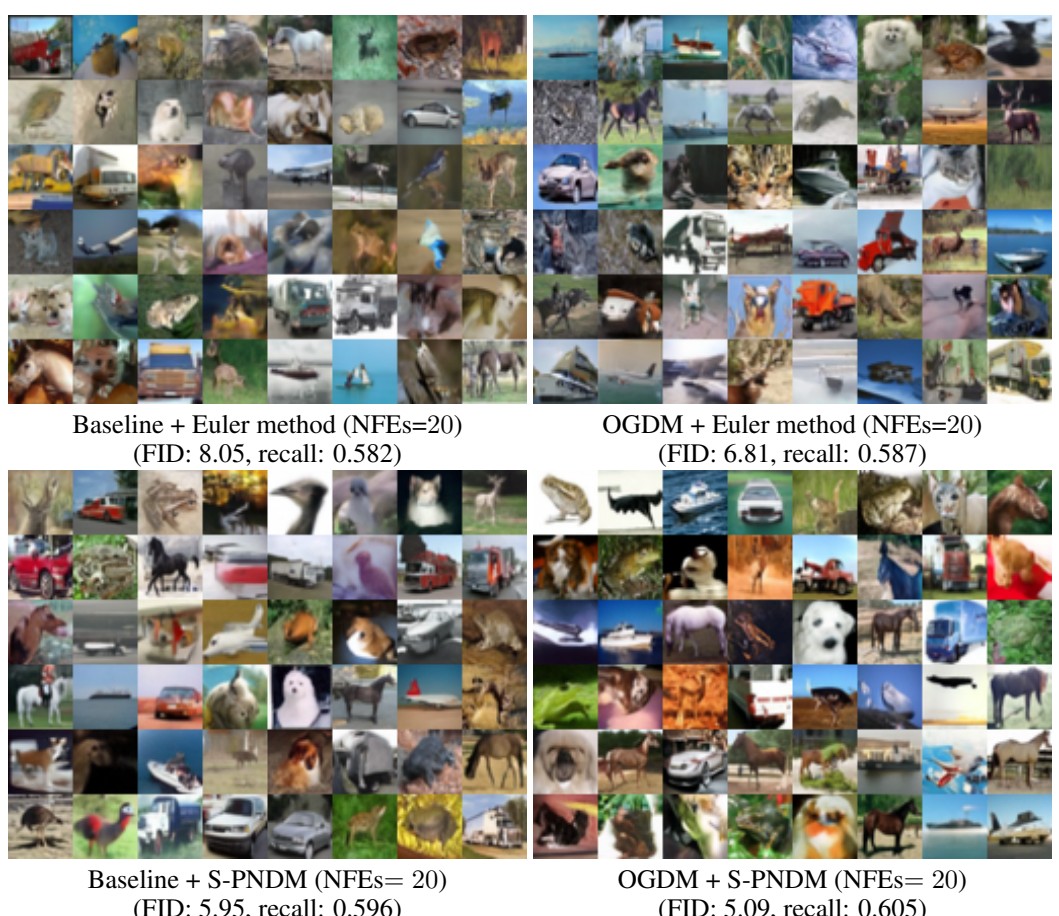

Baseline + Euler method (NFEs=20)
(FID: 8.05, recall: 0.582)

OGDM + Euler method (NFEs=20)
(FID: 6.81, recall: 0.587)

Baseline + S-PNDM (NFEs= 20)
(FID: 5.95, recall: 0.596)

OGDM + S-PNDM (NFEs= 20)
(FID: 5.09, recall: 0.605)

Figure 8: Qualitative results on CIFAR-10 dataset with the ADM backbone using Euler method (top) and S-PNDM (bottom) with NFEs=20.

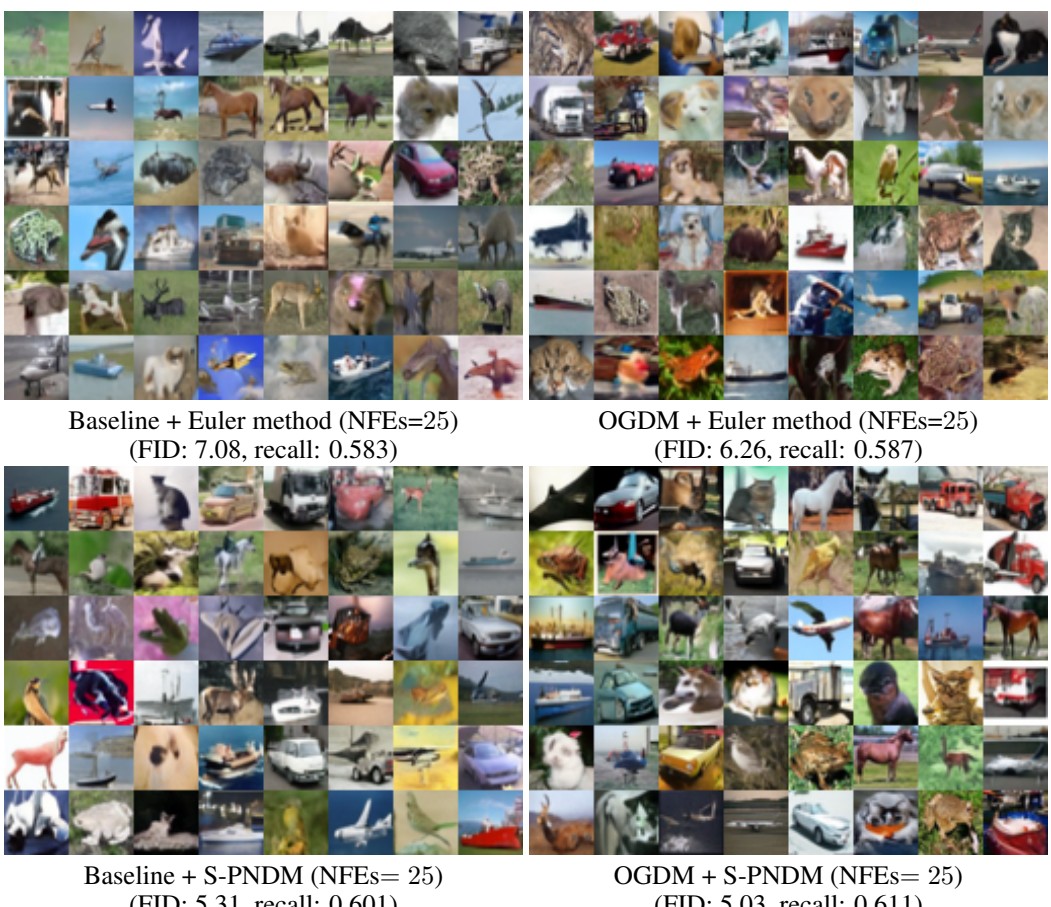

Baseline + Euler method (NFEs=25)
(FID: 7.08, recall: 0.583)

OGDM + Euler method (NFEs=25)
(FID: 6.26, recall: 0.587)

Baseline + S-PNDM (NFEs= 25)
(FID: 5.31, recall: 0.601)

OGDM + S-PNDM (NFEs= 25)
(FID: 5.03, recall: 0.611)

Figure 9: Qualitative results on CIFAR-10 dataset with the ADM backbone using Euler method (top) and S-PNDM (bottom) with NFEs=25.

## E.2 CIFAR-10 SAMPLES WITH EDM BASELINE

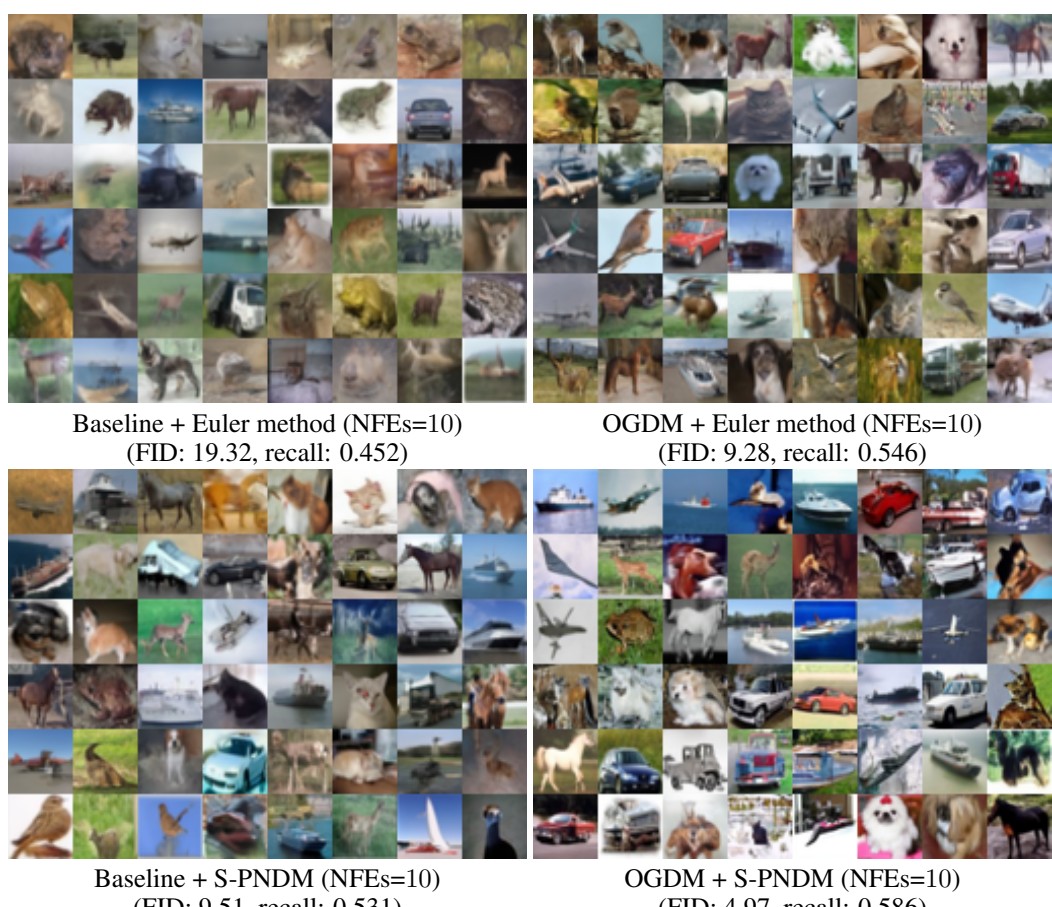

Baseline + Euler method (NFEs=10)
(FID: 19.32, recall: 0.452)

OGDM + Euler method (NFEs=10)
(FID: 9.28, recall: 0.546)

Baseline + S-PNDM (NFEs=10)
(FID: 9.51, recall: 0.531)

OGDM + S-PNDM (NFEs=10)
(FID: 4.97, recall: 0.586)

Figure 10: Qualitative results on CIFAR-10 dataset with the EDM backbone using Euler method (top), and S-PNDM (bottom) with NFEs=10.

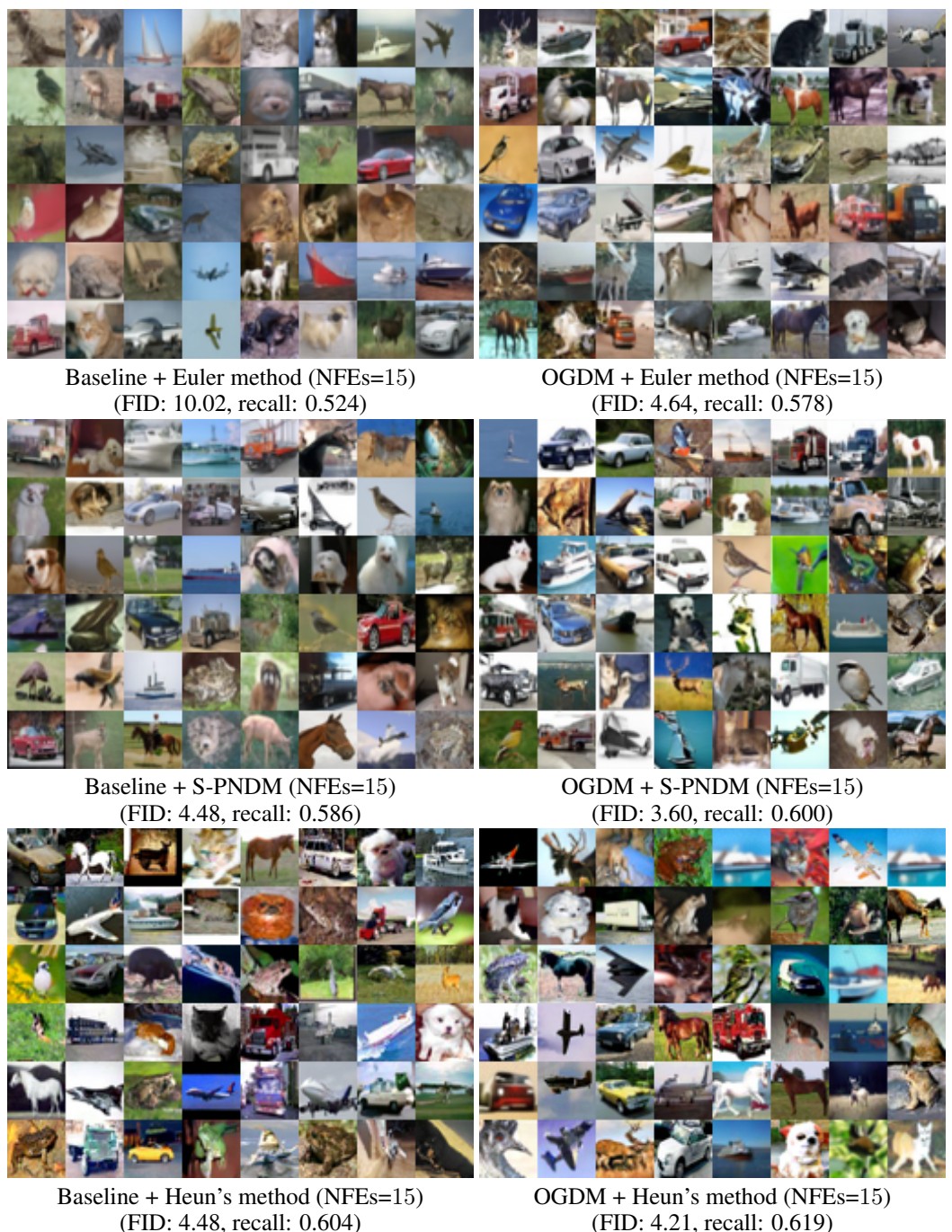

Baseline + Euler method (NFEs=15)
(FID: 10.02, recall: 0.524)

OGDM + Euler method (NFEs=15)
(FID: 4.64, recall: 0.578)

Baseline + S-PNDM (NFEs=15)
(FID: 4.48, recall: 0.586)

OGDM + S-PNDM (NFEs=15)
(FID: 3.60, recall: 0.600)

Baseline + Heun's method (NFEs=15)
(FID: 4.48, recall: 0.604)

OGDM + Heun's method (NFEs=15)
(FID: 4.21, recall: 0.619)

Figure 11: Qualitative results on CIFAR-10 dataset with the EDM backbone using Euler method (top), S-PNDM (middle) and Heun's method (bottom) with NFEs=15.

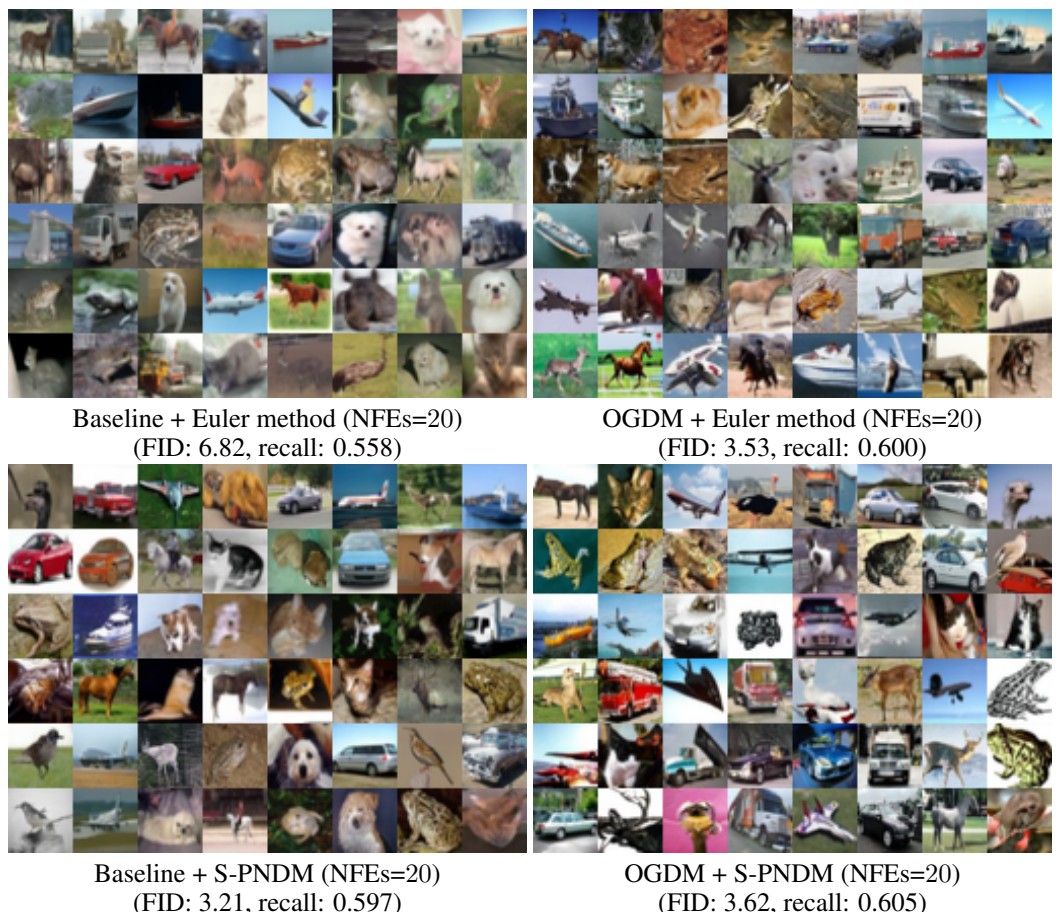

Baseline + Euler method (NFEs=20)
(FID: 6.82, recall: 0.558)

OGDM + Euler method (NFEs=20)
(FID: 3.53, recall: 0.600)

Baseline + S-PNDM (NFEs=20)
(FID: 3.21, recall: 0.597)

OGDM + S-PNDM (NFEs=20)
(FID: 3.62, recall: 0.605)

Figure 12: Qualitative results on CIFAR-10 dataset with the EDM backbone using Euler method (top), and S-PNDM (bottom) with NFEs=20.

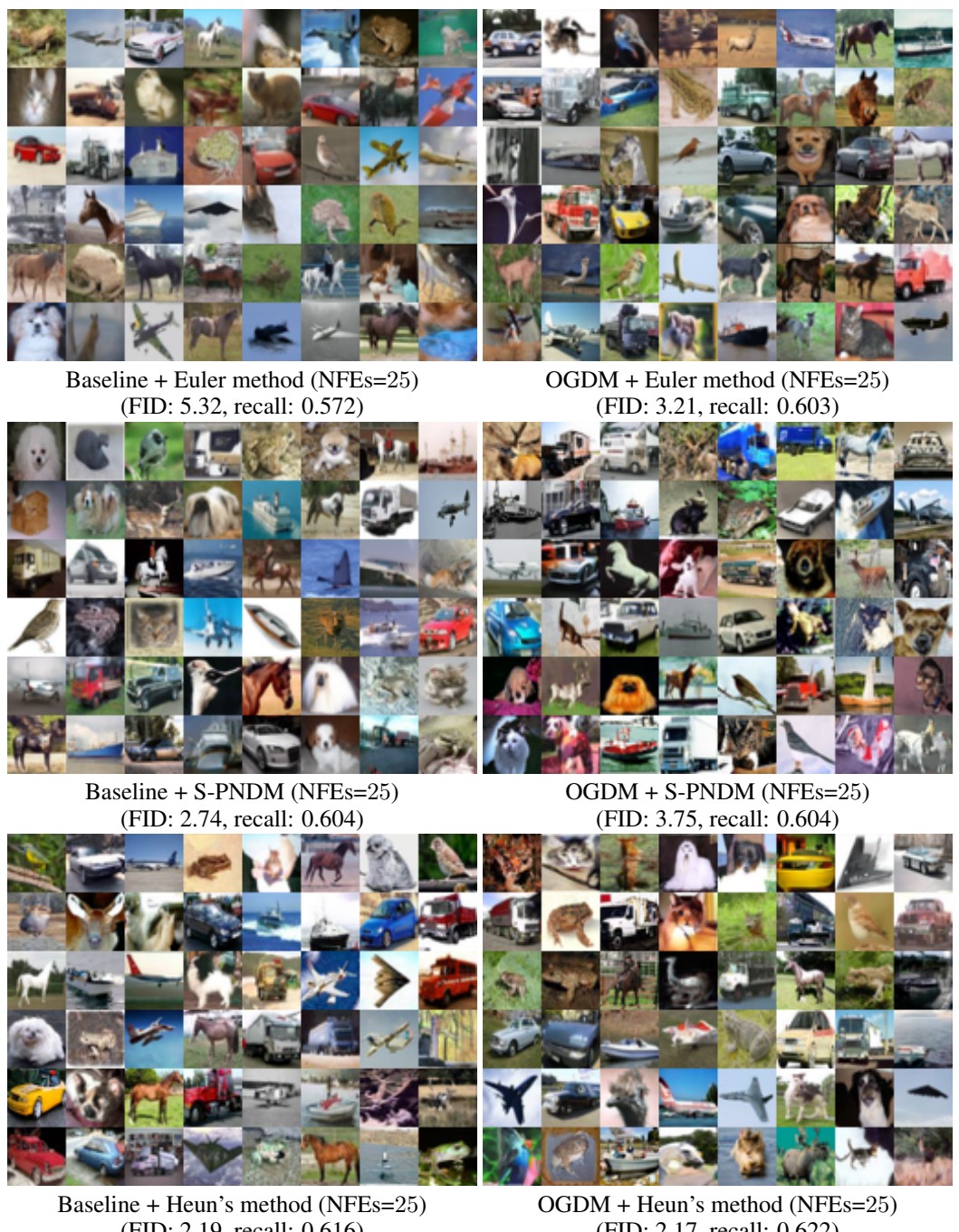

Figure 13: Qualitative results on CIFAR-10 dataset with the EDM backbone using Euler method (top), S-PNDM (middle) and Heun's method (bottom) with NFEs=25.

E.3 CELEBA SAMPLES WITH ADM BASELINE

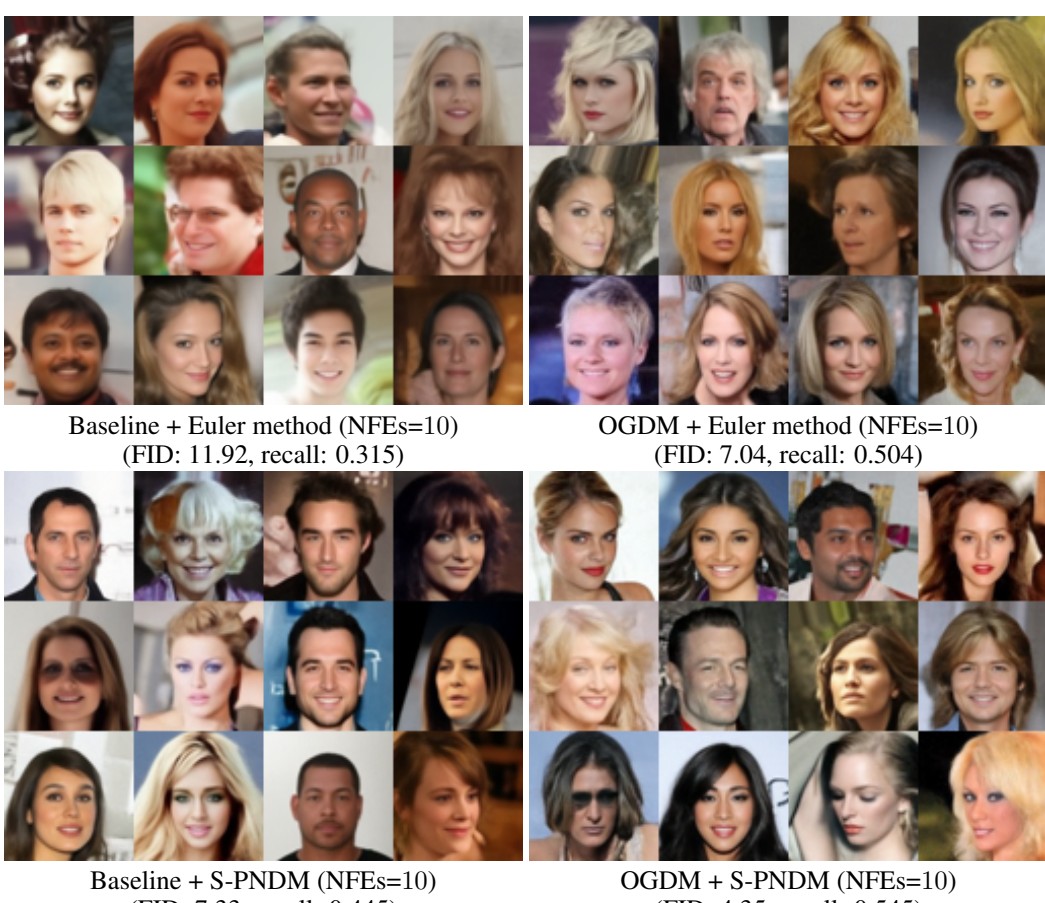

Baseline + Euler method (NFEs=10)
(FID: 11.92, recall: 0.315)

OGDM + Euler method (NFEs=10)
(FID: 7.04, recall: 0.504)

Baseline + S-PNDM (NFEs=10)
(FID: 7.33, recall: 0.445)

OGDM + S-PNDM (NFEs=10)
(FID: 4.35, recall: 0.545)

Figure 14: Qualitative results on CelebA dataset with the ADM backbone using Euler method (top) and S-PNDM (bottom) with NFEs=10.

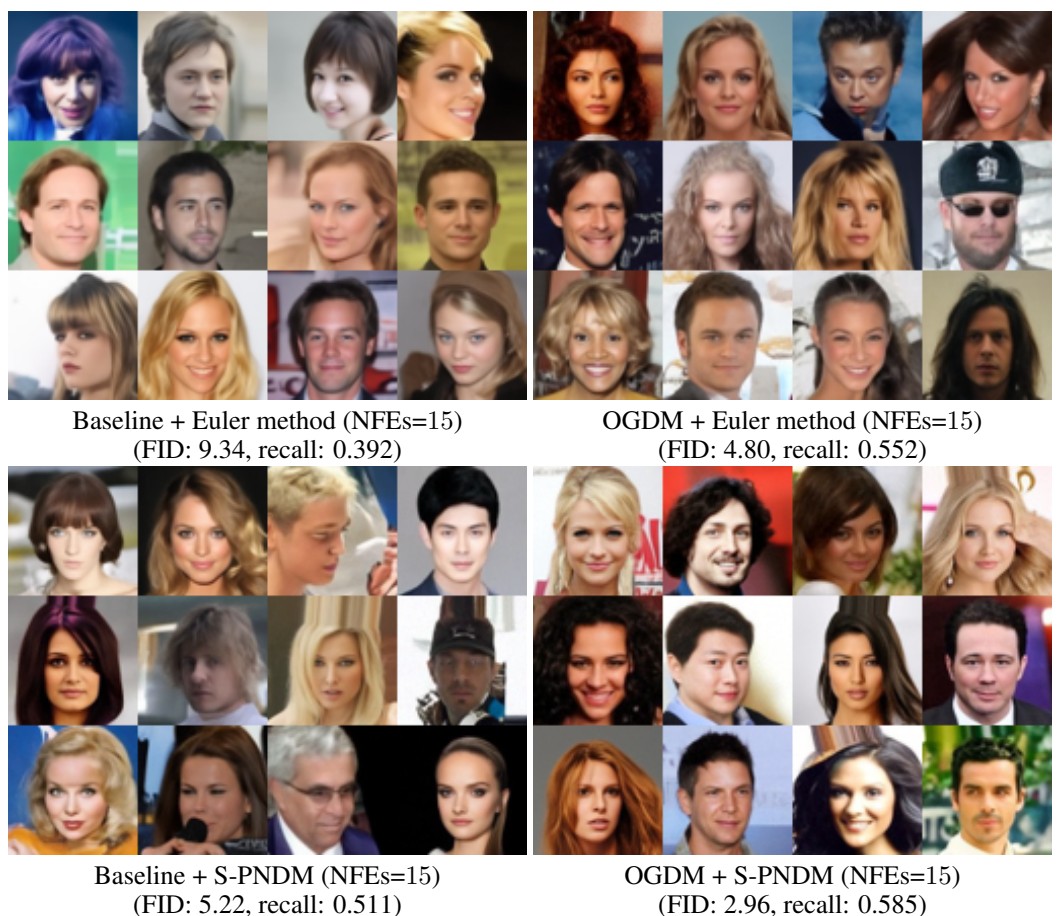

Baseline + Euler method (NFEs=15)
(FID: 9.34, recall: 0.392)

OGDM + Euler method (NFEs=15)
(FID: 4.80, recall: 0.552)

Baseline + S-PNDM (NFEs=15)
(FID: 5.22, recall: 0.511)

OGDM + S-PNDM (NFEs=15)
(FID: 2.96, recall: 0.585)

Figure 15: Qualitative results on CelebA dataset with the ADM backbone using Euler method (top) and S-PNDM (bottom) with NFEs=15.

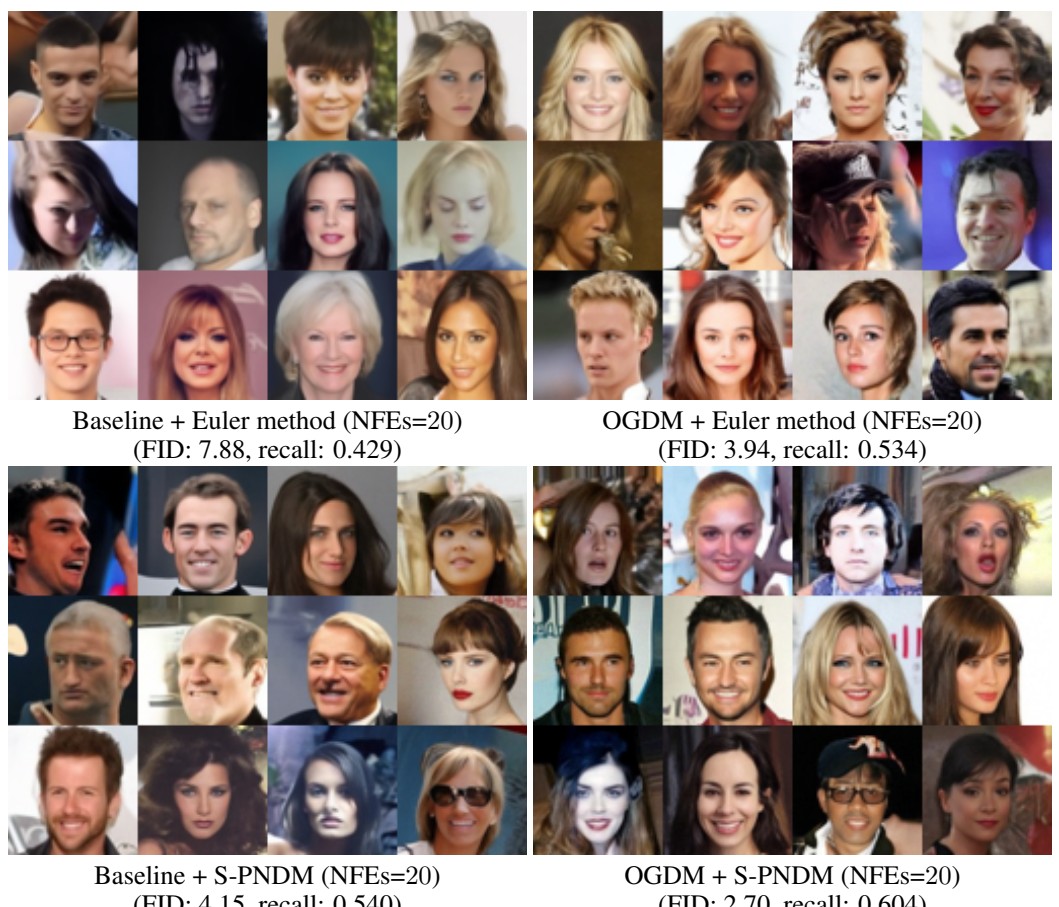



Baseline + Euler method (NFEs=20)
(FID: 7.88, recall: 0.429)
    
OGDM + Euler method (NFEs=20)
(FID: 3.94, recall: 0.534)

Baseline + S-PNDM (NFEs=20)
(FID: 4.15, recall: 0.540)
    
OGDM + S-PNDM (NFEs=20)
(FID: 2.70, recall: 0.604)



Figure 16: Qualitative results on CelebA dataset with the ADM backbone using Euler method (top) and S-PNDM (bottom) with NFEs=20.

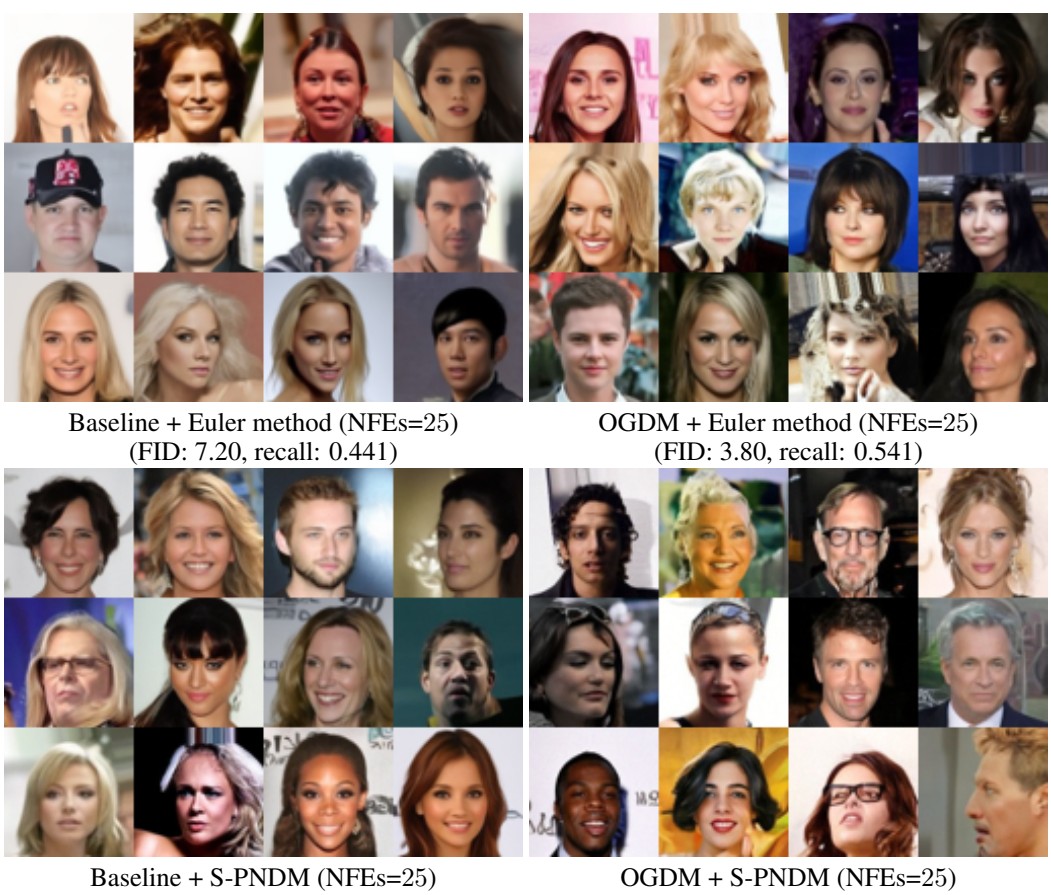

Figure 17: Qualitative results on CelebA dataset with the ADM backbone using Euler method (top) and S-PNDM (bottom) with NFEs=25.

### E.4 LSUN CHURCH SAMPLES WITH LDM BASELINE

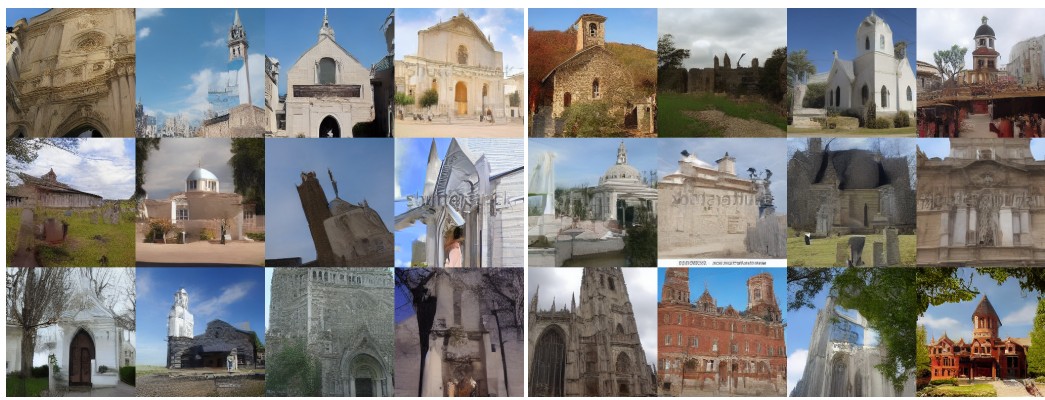



Baseline + Euler method (NFEs=10)      OGDM + Euler method (NFEs=10)
(FID: 15.02, recall: 0.326)           (FID: 14.84, recall: 0.331)



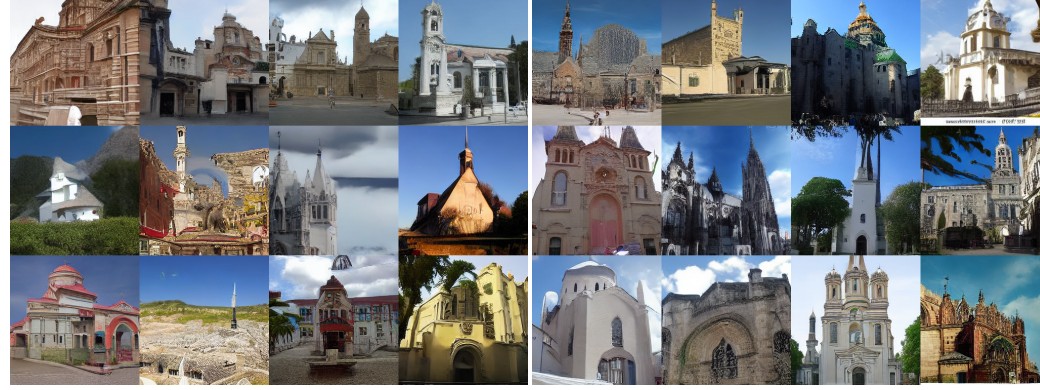



Baseline + S-PNDM (NFEs= 10)      OGDM + S-PNDM (NFEs= 10)
(FID: 9.14, recall: 0.464)          (FID: 8.68, recall: 0.478)



Figure 18: Qualitative results on LSUN Church dataset with the LDM backbone using Euler method (top) and S-PNDM (bottom) with NFEs=10.

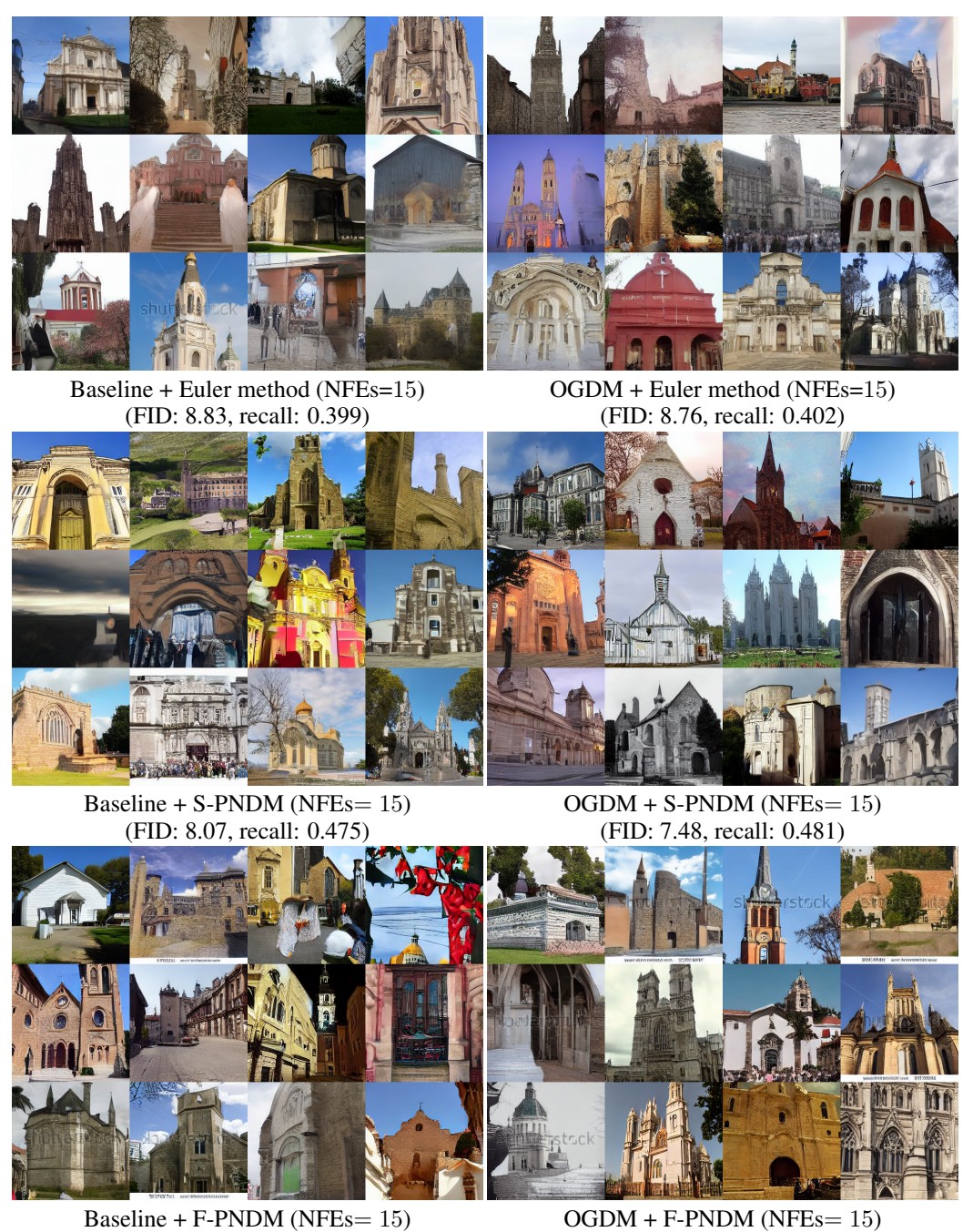

Baseline + Euler method (NFEs=15)
(FID: 8.83, recall: 0.399)

OGDM + Euler method (NFEs=15)
(FID: 8.76, recall: 0.402)

Baseline + S-PNDM (NFEs= 15)
(FID: 8.07, recall: 0.475)

OGDM + S-PNDM (NFEs= 15)
(FID: 7.48, recall: 0.481)

Baseline + F-PNDM (NFEs= 15)
(FID: 12.75, recall: 0.493)

OGDM + F-PNDM (NFEs= 15)
(FID: 11.78, recall: 0.505)

Figure 19: Qualitative results on LSUN Church dataset with the LDM backbone using Euler method (top), S-PNDM (middle) and F-PNDM (bottom) with NFEs=15.

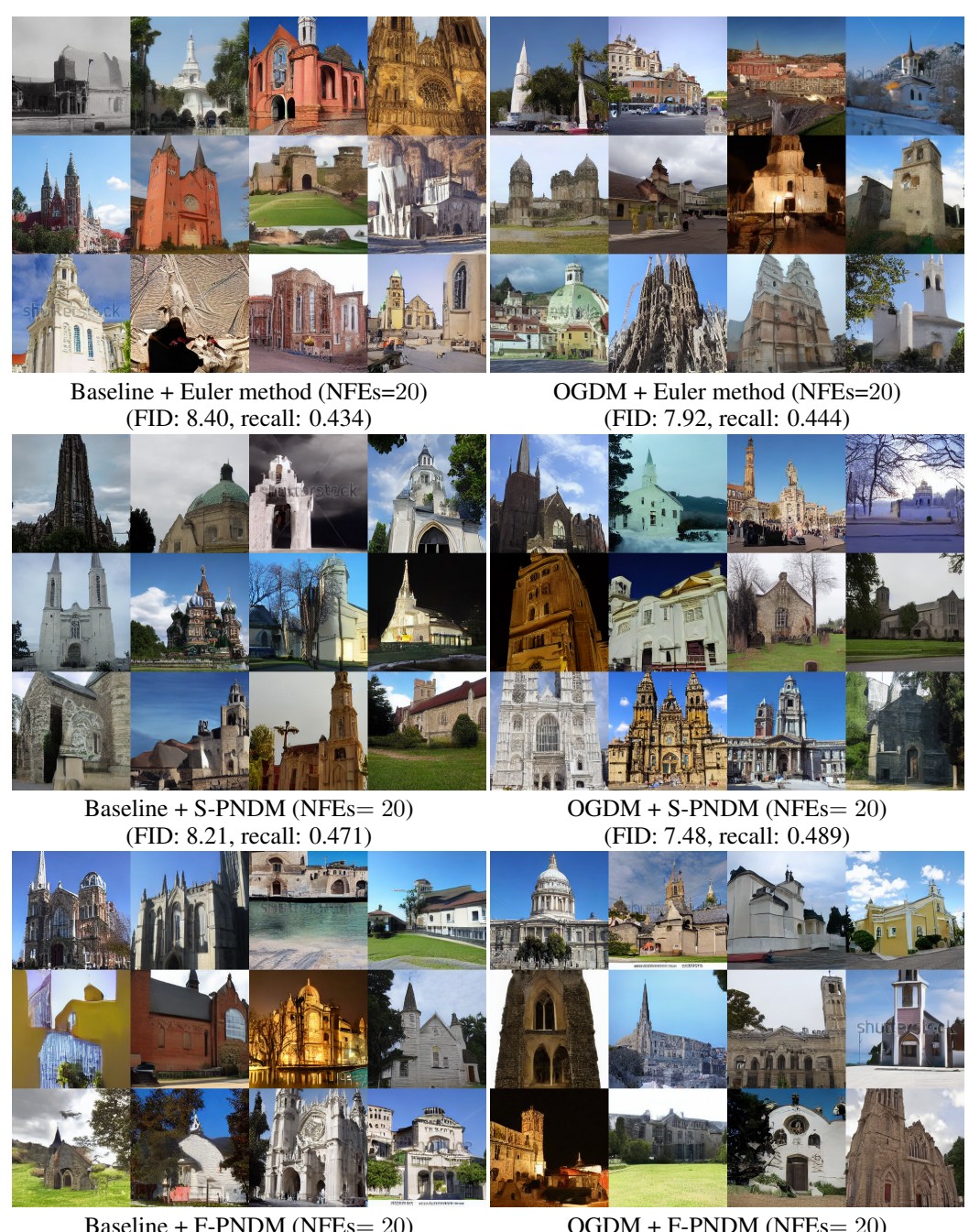

Baseline + Euler method (NFEs=20)
(FID: 8.40, recall: 0.434)

OGDM + Euler method (NFEs=20)
(FID: 7.92, recall: 0.444)

Baseline + S-PNDM (NFEs= 20)
(FID: 8.21, recall: 0.471)

OGDM + S-PNDM (NFEs= 20)
(FID: 7.48, recall: 0.489)

Baseline + F-PNDM (NFEs= 20)
(FID: 9.10, recall: 0.483)

OGDM + F-PNDM (NFEs= 20)
(FID: 8.39, recall: 0.495)

Figure 20: Qualitative results on LSUN Church dataset with the LDM backbone using Euler method (top), S-PNDM (middle) and F-PNDM (bottom) with NFEs=20.

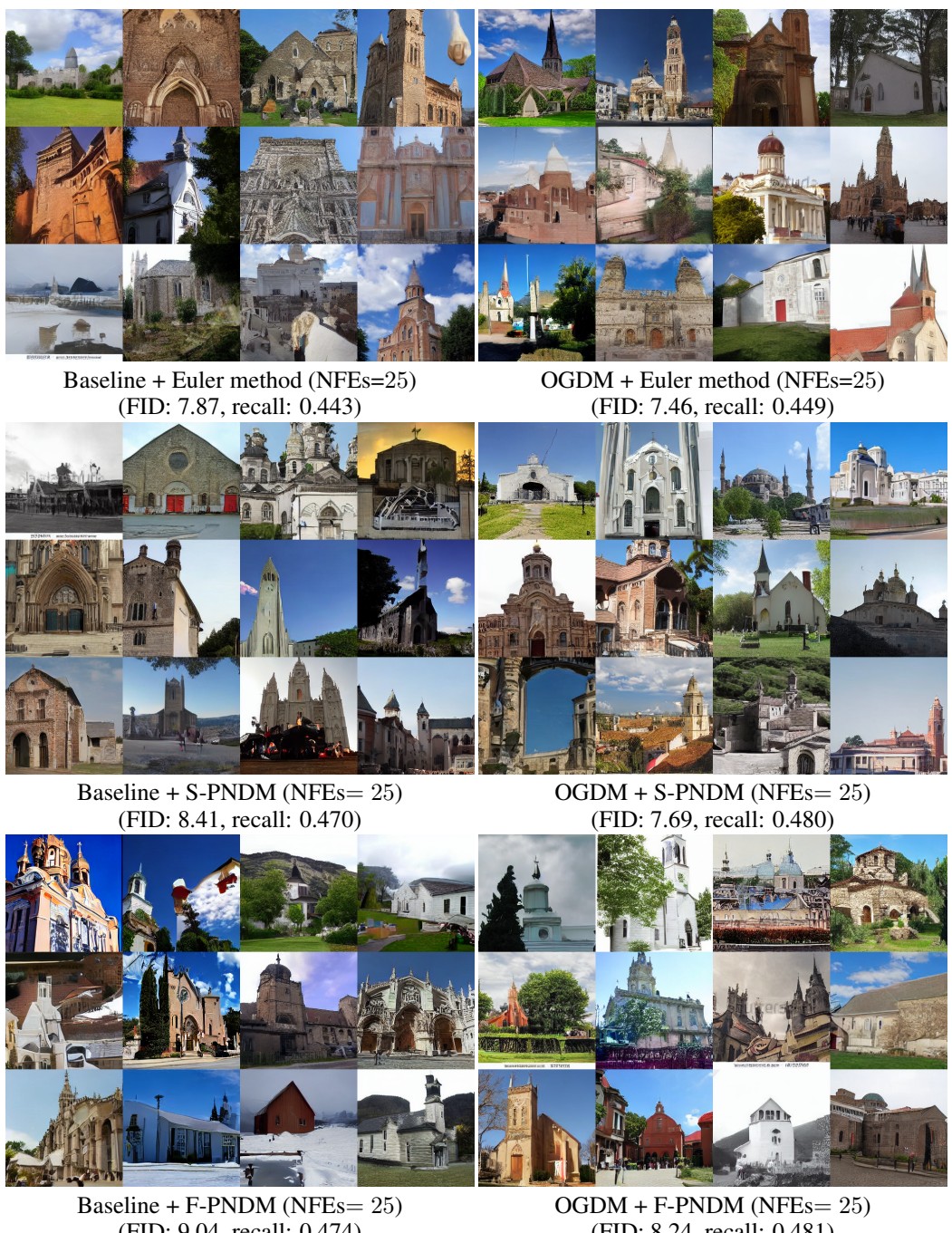

Baseline + Euler method (NFEs=25)
(FID: 7.87, recall: 0.443)

OGDM + Euler method (NFEs=25)
(FID: 7.46, recall: 0.449)

Baseline + S-PNDM (NFEs= 25)
(FID: 8.41, recall: 0.470)

OGDM + S-PNDM (NFEs= 25)
(FID: 7.69, recall: 0.480)

Baseline + F-PNDM (NFEs= 25)
(FID: 9.04, recall: 0.474)

OGDM + F-PNDM (NFEs= 25)
(FID: 8.24, recall: 0.481)

Figure 21: Qualitative results on LSUN Church dataset with the LDM backbone using Euler method (top), S-PNDM (middle) and F-PNDM (bottom) with NFEs=25.

