# OpenReview forum: "Observation-Guided Diffusion Probabilistic Models"
_ICLR.cc/2024/Conference — ICLR 2024 Conference Withdrawn Submission_

### Official Review · Reviewer_oGQD · 2023-10-31

**Soundness:** 2 fair
**Presentation:** 2 fair
**Contribution:** 2 fair
**Rating:** 3
**Confidence:** 4

**Summary:**

This paper proposes observation-guided diffusion probabilistic model (OGDM), which essentially incorporates an additional GAN component that tries to further match the simulated (via ODE) distribution and true (noise injection) distribution. The proposed method accelerates inference speed while maintaining high sample quality by altering only the training procedures of diffusion models. Empirically, it demonstrates benefits across datasets and diffusion baselines.

**Strengths:**

* The authors' proposed method OGDM is novel to my knowledge. There were existing works that also explored utilizing a discriminator to help diffusion model training/sampling, but this paper's proposal is different from those.
* Empirically, the proposed method demonstrates nontrivial benefits when incorporated into various baselines, especially with few NFEs.

**Weaknesses:**

1. There is no empirical comparison between the proposed method and the existing diffusion works that utilize a discriminator. Currently, the paper only compares with the "vanilla" diffusion training baselines. As we have already seen from prior works, incorporating a GAN component into diffusion models could improve the empirical results by a lot, I would highly suggest the authors do so, e.g. [1], [2], to give a clear picture of where this method stands against its peers.
2. The presentation of the paper feels unnecessarily complicated at times. In my opinion, for example, Fig.2, Sec.3.1, and 3.2 are quite distant from the presented method in Sec.3.4. $y$ which at first is defined to be some observation of the underlying diffusion process, but only to materialize as true or fake label. I suggest the authors make the story more straightforward, like the related work [1] and [2].
3. I find Sec.3.5 lacks motivation and clarity, and does not contribute to the overall paper. Essentially the whole subsection is built upon an unreal assumption: for any $\beta$, $p_{u|v}$ can be approximated by some weighted geometric mean of two distributions. The authors claim to provide validity of the approximation, but only to analyze a toy example where the base distribution is Gaussian.


[1] Xiao et al. "Tackling the generative learning trilemma with denoising diffusion gans." ICLR 2022.

[2] Kim et al. "Refining generative process with discriminator guidance in score-based diffusion models." ICML 2023.

**Questions:**

1. In the introduction, the paper talks about diffusion models with DDPM's formulation with "thousands of NFEs" and "reverse Gaussian assumption", but only presents experiments with fast deterministic samplers, where these issues do not arise. Related to this: in Sec.3.5, are you suggesting you are working with a stochastic sampling process that is fast? (You mentioned fast sampling, and also each reverse step is defined as a Gaussian so I assume so.) This is not standard practice nor what is considered in the experiments.
2. In Eq.18, how do you get $x_{t-s}$? Just add a smaller noise to data, which corresponds to one with time step $t-s$?
3. For Eq.25 and the associated explanation, I do not think your method considers the second KL term, but rather it considers the KL between the marginals of $p$ and $q$ at time $\tau_{i-1}$. In other words, the discriminator is not aware of $x_{\tau_i}$ in your formulation.
4. Could you explain why the discriminator needs both $t$ and $s$ as input, rather than just $t-s$? If $t_1-s_1 = t_2-s_2$, then $q(x_{t_1-s_1})$ and $q(x_{t_2-s_2})$ should have the same marginal, no?
5. I suggest the authors provide training overhead (if trained from scratch) of the proposed method.

---

> ### Author Response · Authors · 2023-11-13
> **Responses to Weaknesses**
>
> Thank you for your review. Please read our answers and let us know if you have any further issues or questions.
>
> **1. Response to W1**
>
> We compare our method with DDGAN in the table below. Ours achieve better FID and recall scores although DDGAN requires fewer steps. Note that the performance of DDGAN tends to drop with increasing time steps at some point (NFEs$>4$) which limits the improvement of the model at the expense of the sampling cost.
>
> |  NFEs |      | 20     |      | 8      |      | 4      |      | 2      |       | 1      |
> |:-----:|:----:|--------|:----:|--------|:----:|--------|:----:|--------|:-----:|--------|
> |       |  FID | recall |  FID | recall |  FID | recall |  FID | recall |  FID  | recall |
> | DDGAN |   -  |    -   | 4.36 |  0.56  | 3.75 |  0.57  | 4.08 |  0.54  | 14.60 |  0.19  |
> |  OGDM | 3.53 |  0.60  | 6.16 |  0.58  |   -  |    -   |   -  |    -   |   -   |    -   |
>
> DDGAN is more of a variant of GAN rather than a diffusion model since its objective is not log-likelihood driven. Also, DDGAN does not support deterministic sampling which makes it hard to solve problems such as inversion. For these reasons, we did not perform a separate comparison in the original submission.
> Regarding the comparison with DG[2], please refer to our response \#2 to reviewer RSLy.
>
> [1] Xiao et al. "Tackling the generative learning trilemma with denoising diffusion gans." ICLR 2022.
>
> [2] Kim et al. "Refining generative process with discriminator guidance in score-based diffusion models." ICML 2023.
>
> **2. Response to W2**
>
> The training objective of the diffusion model in Section 3.4 is derived based on the graphical model in Figure 2. The assumptions and equations in Sections 3.1 to 3.3 are essential to derive the objective of the diffusion model. While $y_t$ are arbitrary observations of $x_t$ from Eq. (1)~(9), we provide a possible example of the observations as to whether the state $x_t$ is on the real manifold or not (see Section 3.2). Realizing $y_t$ by Bernoulli distribution is natural for the aforementioned definition.
>
> **3. Response to W3**
>
> Section 3.5 explains how OGDM effectively minimizes the negative log-likelihood at inference, especially with the fast sampling in a theoretical manner. Therefore, Section 3.5 is an important section that contributes to the overall paper.
> Our approximation is realistic and common in practice:  expressing intractable values with tractable values: approximating certain values by the weighted mean of boundary values. For example, the Euler method, a popular integrator, approximates the average value within the integration section by the value at the start of the section. Similarly, Heun’s method approximates the average value within the integration section by the mean of the start and end values of the section. Both integrators approximate the average function values by the arithmetic mean of boundary values. Our approximation of the reverse distribution in Section 3.5 simply replaces the arithmetic mean with the geometric mean, which is a sound approach. Moreover, the toy example provides the validity of the analyses. Contrary to the reviewer oGQD’s claim, the toy example in Appendix A.4 is conducted on non-Gaussian base distribution; it is a multimodal, non-trivial, but tractable distribution. Furthermore, our assumption is only about $\xi(\tilde{\beta})$, unlike the reviewer's comment.

---

> ### Author Response · Authors · 2023-11-13
> **Responses to Questions**
>
> **1. Response to Q1**
>
> The reverse Gaussian assumption holds only when the step size is small, as mentioned in the third paragraph of Section 2. Therefore, the Gaussian assumption no longer holds for the fast samplers, since it is obvious that their step sizes are large.
> The following tables show that OGDM outperforms the vanilla diffusion models even for the stochastic sampling. However, since it is known that using a deterministic sampler performs better than a stochastic sampler for a few-step sampling, we mostly conduct experiments with deterministic samplers.
>
> ***CIFAR-10***
>
> | NFEs |       | 50    |       | 20    |       | 10    |
> |------|:-----:|-------|:-----:|-------|:-----:|-------|
> |      |  FID  |  Rec  |  FID  |  Rec  |  FID  |  Rec  |
> | ADM  | 14.28 | 0.491 | 25.42 | 0.388 | 44.37 | 0.278 |
> | OGDM |  9.94 | 0.524 | 16.84 | 0.450 | 29.70 | 0.359 |
>
> ***CelebA***
>
> | NFEs |       | 50    |       | 20    |       | 10    |
> |------|:-----:|-------|:-----:|-------|:-----:|-------|
> |      |  FID  |  Rec  |  FID  |  Rec  |  FID  |  Rec  |
> | ADM  | 13.51 | 0.312 | 21.00 | 0.181 | 31.09 | 0.079 |
> | OGDM |  9.62 | 0.400 | 15.74 | 0.277 | 24.22 | 0.158 |
>
> **2. Response to Q2**
>
> $x_{t-s}$ is obtained by the forward process given the data as the reviewer oGQD understands.
>
> **3. Response to Q3**
>
> The samples in diffusion models are generated in an iterative way. Therefore, the samples generated by the model are always conditioned on the previous time step samples except for the first step as in Eq. (11). Thus, $p$ is a conditional distribution, not marginal. It is true that the discriminator is not aware of $x_{\tau_i}$ and that is the reason why $q$ is marginal distribution in the second term of Eq. (25).
>
> **4. Response to Q4**
>
> We agree that using $t-s$ instead of $t$ and $s$ will work. However, adding one dimension increases the number of parameters by less than 1% and we just provide raw information to the networks.
>
> **5. Response to Q5**
>
> Please refer to our response \#1 to the reviewer D5aC.

---

### Official Review · Reviewer_RSLy · 2023-11-01

**Soundness:** 3 good
**Presentation:** 3 good
**Contribution:** 2 fair
**Rating:** 3
**Confidence:** 3

**Summary:**

This paper proposes a new training objective for diffusion models. Compared to the original DDPM framework, an additional term is added into the loss function, with the goal of fooling a discriminator to predict the image after one-step denoising as real image. The proposed objective is grounded in a framework where a sequence of observations are added into the original forward and backward process of DDPM. And each observed variable $y_t$ is defined as whether the associated image $x_t$ is from the real image distribution or not. Experiments show that models trained with the proposed objective (both trained from scratch and fine-tuned from pre-trained diffusion models) outperform the baselines in terms of FID and recall, especially when the number of denoising steps is small.

**Strengths:**

1. The paper proposes a new training objective for diffusion models that is theoretically grounded.
2. The paper did comprehensive experiments on three datasets of various resolutions, using various sampling algorithms.

**Weaknesses:**

1. The proposed training pipeline is coupled with a specific sampling algorithm. At inference time, when the sampling algorithm is changed to another one that is different from the one used during training, the proposed method has limited improvements compared to baselines, as demonstrated in Table 3. This limits the applicability of the method, since if new sampling algorithm is proposed, the diffusion model also needs to be re-trained.
2. Comparison with important baselines are missing. Specifically, the discriminator guidance [1] should also be compared, since it also utilizes a discriminator, and it can be applied at inference time without re-training diffusion models. In particular, the reported performance in Table 2 seems to be worse than the number reported in [1].
3. The cost for training diffusion models should also be reported.

[1] Kim et al., 2023. Refining generative process with discriminator guidance in score-based diffusion models.

**Questions:**

1. Why is the projection function $f(\cdot)$ defined as the one-step denoising of the diffusion model?

---

> ### Author Response · Authors · 2023-11-13
> **Responses**
>
> Thank you for your review. Please read our answers and let us know if you have any further issues or questions.
>
> **1. Coupling with a specific sampling algorithm**
>
> Unlike the reviewer RSLy’s comment, the combinations of OGDM trained using Euler projection and PNDMs have shown good performance as in Table 3 and Table 4, especially for S-PNDM with NFEs $\leq 15$. These imply that the coupling of the training pipeline and sampling algorithms is not imperative. We have already discussed this in the second paragraph of Section 4.2. Moreover, most diffusion models adopt the Euler method and the availability of fine-tuning mitigates such concerns.
>
> **2. Comparison with [1]**
>
> We have already mentioned that the motivations and roles of discriminators in [1] and OGDM are different in the last paragraph of Section 2. Moreover, the target problems of [1] and ODGM are completely different; [1] aims for high-quality sampling regardless of the computational cost; OGDM aims for fast sampling with less degradation of sampling quality.
>
>
> | \# steps ($n$)   |         | $13$   |         | $9$    |          |  $7$    |           | $6$     |  Remark                        |
> |----------------------|------:|----------|-------:|---------|--------:|----------|--------:|----------|--------------------------------|
> |                          |  FID | recall  |  FID  | recall |  FID  | recall   |  FID   | recall   |                                      |
> | EDM (Base)     | 2.19 |  0.616 | 3.33 |  0.615 |  7.18 |  0.584 | 15.69 |  0.535 | requires ($2n-1$) NFEs |
> | EDM $+$ [1]   | 1.99 |  0.630 | 4.62 |  0.613 | 12.62 |  0.548 | 24.78 |  0.465 | requires ($3n-2$) NFEs |
> | OGDM              | 2.17 |  0.622 | 2.99 |  0.622 |  6.70 |  0.613 | 13.59 |  0.591 | requires ($2n-1$) NFEs |
> | OGDM $+$ [1] | 2.00 |  0.633 | 3.58 |  0.624 |  8.99 |  0.604 | 19.25 |  0.560 | requires ($3n-2$) NFEs |
>
> As shown in the above table, [1] does not work well for fast sampling. We guess the result originates from the fact that the discriminator in [1] only fixes the training error.
>
> [1] Kim et al., 2023. Refining generative process with discriminator guidance in score-based diffusion models.
>
> **3. Training cost**
>
> Please refer to our response \#1 to reviewer D5aC.
>
> **Response to Q1**
>
> Designing the projection function offers the flexibility in providing some meaningful observations on $x_t$. In OGDM, to accelerate the sampling process, we employ a single discretization step of ODE solvers as a projection function and apply it to time steps $t$ and $t-s$. We utilize well-known ODE solvers such as Euler’s and Heun’s methods, which have proven effective as a sampler for diffusion models and are simple to implement to integrate with training.

---

### Official Review · Reviewer_D5aC · 2023-11-03

**Soundness:** 3 good
**Presentation:** 3 good
**Contribution:** 2 fair
**Rating:** 5
**Confidence:** 4

**Summary:**

This paper proposes a new training method for the diffusion models, where a discriminative loss is designed to reduce the sampling steps for inference. Experiments on several datasets show the proposed method performs better than baselines.

**Strengths:**

1. The paper is well written and clear.
2. The paper presents detailed theoretic analysis for the proposed method.
3. Experiments on several datasets show the effectiveness of the proposed method.

**Weaknesses:**

1. The paper introduces additional cost for training, but there is no additional training cost analysis.
2. The advantage of diffusion models compared to GAN is the training stability, it introduce GAN training again, which may harm the training stability.
3. The experiments are conducted on unconditional generation, leaving its performance on the mainstream text-to-image generation models unclear.

**Questions:**

Please provide the training memory and time cost, and better to provide its performance for Pretrained conditional text-to-image models such as Stable Diffusion.

---

> ### Author Response · Authors · 2023-11-13
> **Responses**
>
> Thank you for your review. Please read our answers and let us know if you have any further issues or questions.
>
> **1. Training cost**
>
> OGDM requires 80% more time and memory for the same iteration of training, but it tends to converge faster than the vanilla diffusion model. For the ADM baseline, vanilla diffusion models score the best FID at 280K iterations while OGDM scores the best FID at 210K. Moreover, the availability of fine-tuning mitigates such concerns requiring less than 10% of the training iterations of baseline training as mentioned in Section 4.2 of the main submission.
>
> **2. Training stability regarding GAN**
>
> The proposed method provides training stability despite the GAN component. One reason for the instabilities in GAN training is the objective of discriminators is usually easier than that of the generators. On the other hand, OGDM provides challenging examples to discriminators preventing it from getting too successful. This is because the denoising network, which works as a generator in terms of GAN training, is guided by the transition loss in Eq. (17), and the target distributions are smoothed by Gaussian perturbation.
>
> **3. T2I generation model**
>
> Training T2I models requires significant computational resources, which is often infeasible, and it is not often covered in many acceleration works. Instead, we conduct experiments with three distinctive datasets using three baselines, and four samplers to demonstrate that our proposed approach is sufficiently generalizable and scalable.